# Genome analysis of Parmales, the sister group of diatoms, reveals the evolutionary specialization of diatoms from phago-mixotrophs to photoautotrophs

Hiroki Ban [1], Shinya Sato[2], Shinya Yoshikawa[2], Kazumasa Yamada [2], Yoji Nakamura [3],
Mutsuo Ichinomiya[4], Naoki Sato[5], Romain Blanc-Mathieu [1,6], Hisashi Endo [1], Akira Kuwata [7✉] &
Hiroyuki Ogata [1✉]

The order Parmales (class Bolidophyceae) is a minor group of pico-sized eukaryotic marine phytoplankton that contains species with cells surrounded by silica plates. Previous studies revealed that Parmales is a member of ochrophytes and sister to diatoms (phylum Bacillariophyta), the most successful phytoplankton group in the modern ocean. Therefore, parmalean genomes can serve as a reference to elucidate both the evolutionary events that differentiated these two lineages and the genomic basis for the ecological success of diatoms vs. the more cryptic lifestyle of parmaleans. Here, we compare the genomes of eight parmaleans and five diatoms to explore their physiological and evolutionary differences. Parmaleans are predicted to be phago-mixotrophs. By contrast, diatoms have lost genes related to phagocytosis, indicating the ecological specialization from phago-mixotrophy to photoautotrophy in their early evolution. Furthermore, diatoms show significant enrichment in gene sets involved in nutrient uptake and metabolism, including iron and silica, in comparison with parmaleans. Overall, our results suggest a strong evolutionary link between the loss of phago-mixotrophy and specialization to a silicified photoautotrophic life stage early in diatom evolution after diverging from the Parmales lineage.

[1] Bioinformatics Center, Institute for Chemical Research, Kyoto University, Gokasho, Uji, Kyoto 611-0011, Japan. [2] Department of Marine Science and Technology, Fukui Prefectural University, 1-1 Gakuen-cho, Obama City, Fukui 917-0003, Japan. [3] Bioinformatics and Biosciences Division, Fisheries Stock Assessment Center, Fisheries Resources Institute, Japan Fisheries Research and Education Agency, 2-12-4 Fuku-ura, Kanazawa, Yokohama, Kanagawa 236-8648, Japan. [4] Prefectural University of Kumamoto, 3-1-100 Tsukide, Kumamoto 862-8502, Japan. [5] Graduate School of Arts and Sciences, University of Tokyo, Komaba, Meguro-ku, Tokyo 153-8902, Japan. [6] Laboratoire de Physiologie Cellulaire & Végétale, CEA, Univ. Grenoble Alpes, CNRS, INRA, IRIG, Grenoble, France. [7] Shiogama field station, Fisheries Resources Institute, Japan Fisheries Research and Education Agency, 3-27-5 Shinhama-cho, Shiogama, Miyagi, Japan. ✉email: akuwata@affrc.go.jp; ogata@kuicr.kyoto-u.ac.jp

The order Parmales (class Bolidophyceae) is a group of pico-sized (2–5 µm) eukaryotic marine phytoplankton with cells surrounded by silicified plates[1]. Parmaleans are widespread in the ocean, from polar to tropical regions, and are relatively abundant in polar and subarctic regions[2,3]. Parmalean sequences are most abundant in the picoplanktonic fraction (0.8–5 µm) of the global ocean metabarcoding data from *Tara* Oceans and represent at most 4% of the sequences of photosynthetic organisms and <1% on average[2]. Currently, only 17 taxa of parmaleans have been described[3,4]. SEM and TEM observations, molecular phylogenetics, and photosynthetic pigment analyses indicated that parmaleans belongs to Bolidophyceae (ochrophytes)[5], which is the sister taxon of diatoms (phylum Bacillariophyta). Bolidophyceae also contains pico-sized photosynthetic naked flagellates (called bolidomonads) that mainly inhabit subtropical waters[6]. Recent phylogenetic analyses using several newly isolated strains revealed that flagellated bolidomonad species belong to the silicified and non-flagellated parmalean genus *Triparma* within Bolidophyceae, suggesting that the *Triparma* life cycle switches between silicified/non-flagellated and naked/flagellated stages[2].

Diatoms are a relatively young group of unicellular eukaryotes that are estimated to have emerged near the Triassic-Jurassic boundary (ca. 200 million years ago[7]). Despite their short evolutionary history, diatoms represent the most successful phytoplankton group in the modern ocean; they are highly diverse up to $10^5$ species[8], and contribute extensively to marine primary production, performing up to 20% of total planetary photosynthesis. Diatoms are thought to be particularly successful in dynamic environments such as upwelling areas, and it has been suggested that their ecological success is supported by traits such as silicified cell wall defense[9] and luxury nutrient uptake[10]. However, despite advances in understanding diatom genomes during the last two decades, the reasons underlying the success of diatoms in modern oceans remain poorly understood. To understand the ecological success of diatoms, characterization of the evolution of physiology-related genes in this taxon is necessary.

Although parmaleans are the closest relatives of diatoms, they show much lower biomass, species diversity, and ecological impact than their sister taxon. The proposed parmalean life cycle, which switches between silicified/non-flagellated and naked/flagellated stages, is similar to the proposed origin of diatoms[2]. Ancestral diatoms were possibly haploid flagellates that formed silicified diploid zygotes[11]. The mitotic division of the zygote might have taken place preferentially to give rise to centric diatoms[12], which is the most ancient diatom group with a diploid vegetative stage producing naked flagellated haploid male gametes for sexual reproduction[13]. Thus, a comparison of parmaleans and diatoms is expected to provide important clues on differences in their ecological strategies and evolutionary paths. To date, only limited genomic data on parmaleans have been available[14], and the genomic features and evolutionary events that led to differences between parmaleans and diatoms have remained unstudied. In this study, we generated seven novel parmalean genome assemblies. These seven draft genomes, one previously determined parmalean genome, and five publicly available diatom genomes were used to perform a comparative genome analysis. Our results delineate the evolutionary trajectories of these two lineages after their divergence and correlate their ecological features with their genomic functions.

## Results and discussion
### General genomic features
In this study, we obtained whole-genome sequences of seven parmaleans, including six strains from two genera (*Triparma* and *Tetraparma*) that are frequently

observed in the subarctic Pacific Ocean[4,15], as well as one strain (named 'Scaly parma') from an undescribed taxon that is phylogenetically and morphologically distinct from known parmaleans. Together with the previously sequenced *Triparma laevis* f. *inornata* genome[14], we built a database of eight parmalean strain genomes. Phylogenetic analysis of 18 S rRNA sequences of parmaleans shows our genomes cover the wide range of parmalean group from the most basal clade I ('Scaly parma') to clade III (*Tetraparma*) and IV (*Triparma*) (Supplementary Fig. 1). The parmalean genomes were similar in size, ranging from 31.0 Mb for 'Scaly parma' to 43.6 Mb for *Tetraparma gracilis* (Table 1). The predicted numbers of genes ranged from 12,177 for 'Scaly parma' to 16,002 for *Triparma laevis* f. *longispina* (Table 1). These genome sizes are relatively constant compared to diatom genomes and similar to those of *Thalassiosira pseudonana* (32.4 Mb)[16] and *Phaeodactylum tricornutum* (27.4 Mb)[17], which have rather small genomes among diatoms.

We grouped the genes from the parmaleans (8 strains), diatoms (5 strains), and other stramenopiles (5 strains) and revealed 62,344 of orthologous groups (OGs) including singletons. Phylogenomic analysis based on 175 single-copy OGs among them clearly shows parmaleans are monophyletic and sister to diatoms (Fig. 1a). 34,299 OGs were present only in diatoms or parmaleans and not in other stramenopiles (Fig. 1b: yellow + orange + purple + green in diatoms and Parmales). Of those, only 1,457 OGs were shared by diatoms and parmaleans (Fig. 1b: yellow). 20,957 OGs were specific to diatoms (diatom-specific OGs, Fig. 1b: orange + green in diatoms), and 11,885 OGs were specific to parmaleans (Parmales-specific OGs, Fig. 1b: purple and green in Parmales). 55.1 % of the genes in the core OGs conserved in all analysed strains (1,154 OGs, Fig. 1b: red) had InterPro domains, and 51.1 % of the genes in the OGs shared only by diatoms and parmaleans (1,457 OGs, Fig. 1b: yellow) had InterPro domains. By contrast, only 16.2 % of genes in diatom-specific OGs (20,957 OGs, Fig. 1b: orange + green in diatoms) and 43.5 % of genes in parmalean-specific OGs (11,885 OGs, Fig. 1b: purple and green in Parmales) had InterPro domains.

**Differentially enriched protein domains.** By comparing the eight parmalean and five diatom genomes, we found 60 and 319 InterPro domains in which the diatom and Parmales lineages, respectively, were significantly enriched (Supplementary Data 2, Supplementary Data 3). We noted that diatoms were enriched in cyclin domains and heat-shock transcription factor domains compared to Parmales, consistent with previous data that diatoms contain greater numbers of these proteins than other eukaryotes[16,17] (Fig. 2a). In addition, diatoms were enriched in protease domains and sulfotransferase domains relative to Parmales. Proteases and metalloproteases are known to be induced by limitations of nitrogen, iron, and light[18,19]. Sulfotransferases are enzymes that catalyses sulfonation and are implicated in programmed cell death in *Skeletonema marinoi*, a bloom-forming marine diatom[20]. These gene families are thought to be involved in the stress response process in diatoms.

InterPro domains in which parmaleans were enriched included those involved in intracellular signalling pathways, such as the G protein signalling, cyclic nucleotide signalling, calcium signalling, and action potential pathways (Fig. 2a). G protein-coupled receptors were involved in responses to sexual cues in the planktonic diatom *Pseudo-nitzschia multistriata*[21], and to colonization in the benthic morphotype of *Phaeodactylum tricornutum*[22], that are also known to have two planktonic morphotypes. Diatoms also exhibit action potential signalling to modulate their cellular motility[23,24]. Furthermore, parmaleans encoded a strikingly greater number of calcium-binding proteins

**Table 1 Assembly and annotation results and statistics.**

| | Triparma laevis f. inornata | Triparma laevis f. longispina | Triparma verrucosa | Triparma strigata | Triparma retinervis | Triparma columacea | Tetraparma gracilis | Scaly parma |
|---|---|---|---|---|---|---|---|---|
| Genome size (Mbp) | 42.6 | 41.4 | 35.5 | 35.2 | 36.5 | 43.0 | 43.6 | 31.0 |
| No. of scaffolds | 902 | 1,055 | 659 | 634 | 8,760 | 1,858 | 7,082 | 1,921 |
| N50 (kbp) | 83.2 | 77.9 | 74.0 | 73.4 | 8.2 | 63.5 | 10.7 | 51.7 |
| GC (%) | 49.8 | 51.1 | 52.1 | 52.2 | 52.4 | 51.0 | 64.8 | 51.0 |
| No. of predicted protein-coding genes | 13,396 | 16,002 | 14,488 | 14,364 | 13,636 | 13,919 | 15,310 | 12,177 |
| BUSCO Complete genes (%) | 74 | 93 | 95 | 94 | 68 | 90 | 64 | 91 |

(close to 300) that could act as messenger molecules[25] (Fig. 2b). Intercellular signalling pathways in parmaleans may also be used to sense the external environment similarly to diatoms. The enrichment of these pathways may relate to the putative alternating life cycle stages (i.e., silicified/non-flagellated and naked/flagellated cell stages[2]) of parmaleans, and/or to flagellar movement in response to the environment.

Parmalean genomes were notably enriched in domains associated with lipids and fatty acids (Fig. 2a). For example, diacylglycerol acyltransferase is an enzyme for the terminal step in the production of triacylglycerol, the main component of stored lipids[26]. The steroidogenic acute regulatory protein-related lipid transfer (START) domain that binds to lipids and sterols[27] is one of the domains in which parmalean genomes are most enriched, with up to 141 genes in *Tetraparma gracilis*. This domain sometimes consists of multi-domain proteins and works in lipid trafficking, lipid metabolism, and cell signalling in animals and land plants[27]. START domain-containing proteins in parmaleans also contain other functional domains, such as lipid metabolism enzymes, transporters, kinases, and transcription factors (Fig. 2c). These results suggest diverse lipid-related physiological processes in parmaleans.

**Phagotrophy**. Some InterPro domains in which parmaleans are enriched are known to be involved in phagotrophy[28], including cell adhesion[29], intercellular signalling[30], cytoskeleton[31], lysosome[32], and WASH[33] (WASP and SCAR homolog) complex proteins (Fig. 3a). Using a gene-based phago-mixotrophy prediction model[28], parmaleans were predicted as phago-mixotrophs (high scores > 0.99), whereas diatoms were not (low scores < 0.07) (Fig. 3b). This result suggests that parmaleans are capable of phagocytosis. We also applied this prediction model to the bolidomonads (naked/flagellated parmaleans) transcriptomes, and bolidomonads were also predicted as phago-mixotrophs (high scores >0.96, Supplementary Fig. 2). Although there is no experimental evidence of phagocytosis in silicified parmaleans, field studies demonstrated that bolidomonads feed on cyanobacteria[34,35]. As transcriptome data reflect gene repertoires expressed under specific physiological conditions, bolidomonads might be phagotrophs. It remains unclear which life cycle stages of the parmaleans that we analysed are phagotrophs. However, assuming that bolidomonads indeed represent a part of the parmalean life cycle[3], and a possibility that the silicified parmalean cell wall could physically interfere with feeding bacteria, it is likely that parmaleans perform phagocytosis in their putative naked/flagellated stage (Fig. 3c).

In the following sections, we move from the analysis of enriched domains to more focused investigation of genes in specific pathways and functions.

**Flagellum**. To investigate the possibility that parmaleans can produce a flagellated cell[2], we searched for genes responsible for flagellar motility in the parmalean and diatom genomes and bolidomonad transcriptomes. The searched gene set included intraflagellar transport (IFT) subunit genes[36] of IFT-A complex (6 genes), IFT-B complex (15 genes) and Bardet–Biedl Syndrome proteins (BBSome; 7 genes). Flagellum structural genes for tubulin, radial spokes, dynein arms, and the central pair complex were excluded from analysis because these genes are also involved in other processes/structures (such as the centriole in *Triparma laevis*[37]) and are not unique to the flagellum. For this analysis, bolidomonad transcriptomes and centric diatom genomes were considered as positive controls because of the presence of the flagellar structure[6] and the presence of flagellated sperm in their life cycle[38], respectively. Similarly, pennate diatom genomes were

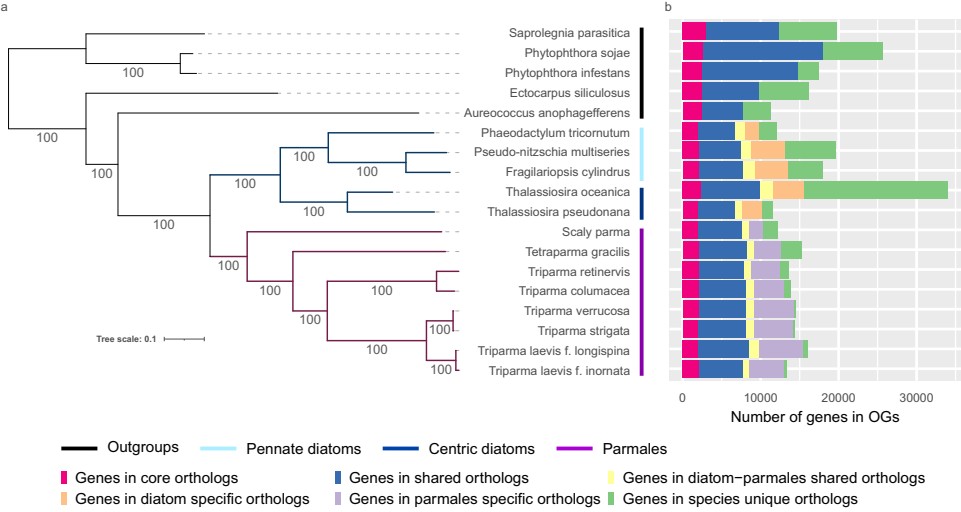

**Fig. 1 Phylogenetic relationships of diatoms, parmaleans (Parmales), and stramenopiles, and number of shared genes in OGs. a** Maximum likelihood tree estimated by RAxML with 175 single-copy OGs. Blue and purple branches are diatom and Parmales clades, respectively. The numbers on the branches represent bootstrap values. The coloured bars indicate the group where each taxon belongs (black: outgroup, light blue: pennate diatoms, deep blue: centric diatoms, purple: Parmales). **b** The barplot represents the number of genes in different categories of orthologous groups. Source data are provided as Supplementary Data 1.

considered as negative controls because flagellar structures have never been observed in this group, even in known sexually reproductive species[39].

A nearly-full set of the flagellar genes were found in parmalean genomes, bolidomonad transcriptomes and other genomes (*Aureococcus anophagefferens* and *Ectocarpus siliculosus*), whereas IFT-A and BBsome genes were completely absent in both types of diatoms (Fig. 3d). IFT-B genes were partially detected in centric diatoms and completely lost in pennate diatoms. These results suggest that parmaleans have a flagellated stage in their life cycle and are consistent with the idea that parmaleans are phago-mixotrophs in their putative naked/flagellated stage. Jensen et al.[40] speculated that the two central microtubules were dispensed within the sperms of centric diatoms. Given the detection of the nearly-full set of flagellar genes in the parmaleans vs. the complete lack of IFT-A and BBSome and partial loss of IFT-B in the centric diatoms, it is possible that evolutionary pressure to maintain the flagellated stage is higher in parmaleans than in centric diatoms. This may be due to the presence of a frequent or prolonged flagellated stage in parmaleans, which is not expected for the sperms of centric diatoms.

**Nitrogen metabolism.** The number of transporter genes involved in the uptake of nitrogen sources differed greatly between diatoms and parmaleans (Fig. 4a). Parmaleans had 0–3 nitrate/nitrite transporter genes, whereas diatoms had 3–7. Only one or no urea transporter gene was detected in each parmalean, whereas 3–6 genes were detected in each diatom. Diatoms tended to have more ammonium transporter genes than parmaleans, although the difference was not as obvious as for the other transporters (2–8 genes for parmaleans vs. 4–10 for diatoms). Vacuolar nitrate transporters, which store nitrogen sources in the vacuole[41] and are considered important for the luxury nutrient uptake of diatoms[42–44], were absent from parmalean genomes. This suggests that parmaleans may be less competent to store nitrogen sources than diatoms, although it remains to be determined if parmaleans utilise another vacuolar nitrate transporter that is not orthologous to that of diatoms.

Parmaleans had all of the ornithine–urea cycle genes, as with diatoms[16] and other stramenopiles[45] (Fig. 4b, Supplementary

Fig. 3). Other involved genes (i.e., those encoding NAD(P)H nitrite reductase, carbamate kinase, formamidase, cyanate lyase, and hydroxylamine reductase) were present in diatoms but absent from parmaleans. NAD(P)H nitrite reductase is a major enzyme in nitrogen metabolism that catalyses the production of ammonium from nitrite. Carbamate kinase is a major enzyme that produces carbamoyl phosphate, which is a precursor of the urea cycle. It should be noted that while parmalean genomes lacked NAD(P)H nitrite reductase, they retained a ferredoxin-nitrite reductase gene also found in diatoms that can perform the same activity. Likewise, parmaleans and diatoms share a carbamoyl phosphate synthetase enzyme that can function in lieu of carbamate kinase (Supplementary Fig. 4). The presence of multiple alternative pathways for these activities in diatoms, as opposed to only one in parmaleans, may enhance the efficiency of their nitrogen metabolism. Formamidase, cyanate lyase, and hydroxylamine reductase function around the main pathway of nitrogen metabolism. Previous studies showed that formamidase and cyanate lyase are upregulated under N-limited conditions in the diatom *Phaeodactylum tricornutum*[46] as well as other ochrophytes such as *Aureococcus anophagefferens*[47]. Diatoms encoding these enzymes may have the ability to obtain ammonium from intercellular nitrogen compounds even when they cannot obtain extracellular nitrogen[46,47]. By contrast, parmaleans lacking these enzymes may not have this capacity.

**Iron metabolism.** Iron acts as an electron carrier in the photosynthesis and multiple other metabolic activities associated with phototrophy. In marine ecosystems, iron is one of the prime limiting elements for phototrophs because of high demand[48]. Therefore, iron uptake ability is an important factor for competition in marine environments. We searched for iron metabolism-related genes in diatom and parmalean genomes. Ferric reductase (FRE), a high-affinity reductive iron uptake system component, was found in all diatoms and parmaleans investigated (Fig. 5a), but parmaleans completely lacked $Fe^{3+}$ permease (FTR) genes (Fig. 5a). Parmalean genomes encoded genes with high sequence similarity to diatom FTR genes, but the parmalean sequences lacked the [REXXE] motif, which is important for iron permeation[49]. This indicates that the

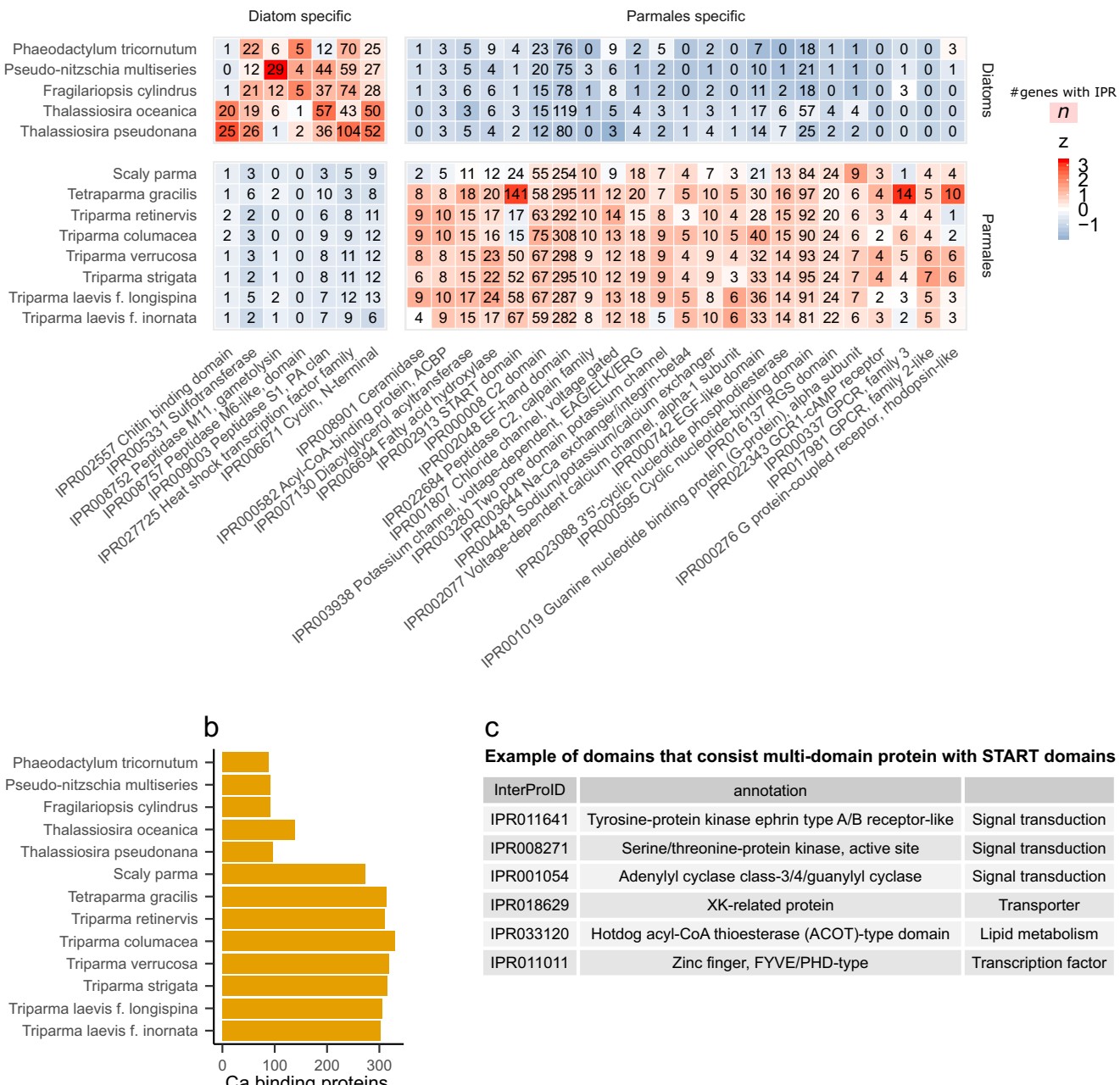

**Fig. 2 Clade enriched gene families. a** InterPro domains enriched in diatom and parmalean genomes. The colours are scaled in ascending order from blue to red by the z-value in each row. Source data are provided as Supplementary Data 2 and Supplementary Data 3. **b** Number of genes annotated with GO:0005509 (calcium ion binding) by InterProScan. Source data are provided as Supplementary Data 5. **c** An example of InterPro domains composed of multi-domain proteins including START domains. Source data are provided as Supplementary Data 4.

diatom/Parmales common ancestor possessed FTR but parmalean FTR homologs may have lost their ability to enable iron permeation during evolution. As for the candidate genes involved in the non-reductive iron uptake system, iron starvation-induced protein 2A (ISIP2A/FEA)[50] was widely distributed in parmaleans, whereas ISIP1 was not present (Fig. 5a). ISIP1 plays an important role in siderophore uptake in diatoms and is considered a highly efficient iron uptake gene[51]. Our results support the idea that ISIP1 is diatom-specific (although there is a report on the possible presence of homologs in some species of pelagophytes, haptophytes, and

dinoflagellates of the genus *Karenia*)[51] and its presence may underlie diatoms' high iron uptake capacity.

Most parmaleans encoded genes for plastocyanin, a copper-containing redox protein that can substitute for cytochrome $c_6$, which is a redox protein that requires iron and transfers electrons from the cytochrome $b_6$–$f$ complex to photosystem I during photosynthesis. It was generally thought that chlorophyll $c$-containing algae lack plastocyanin, but several pelagic diatoms from different genera (including *Thalassiosira oceanica*) encode plastocyanin and are thought to be adapted to iron-deficient pelagic regions[52,53]. Parmaleans may also have an environment-dependent

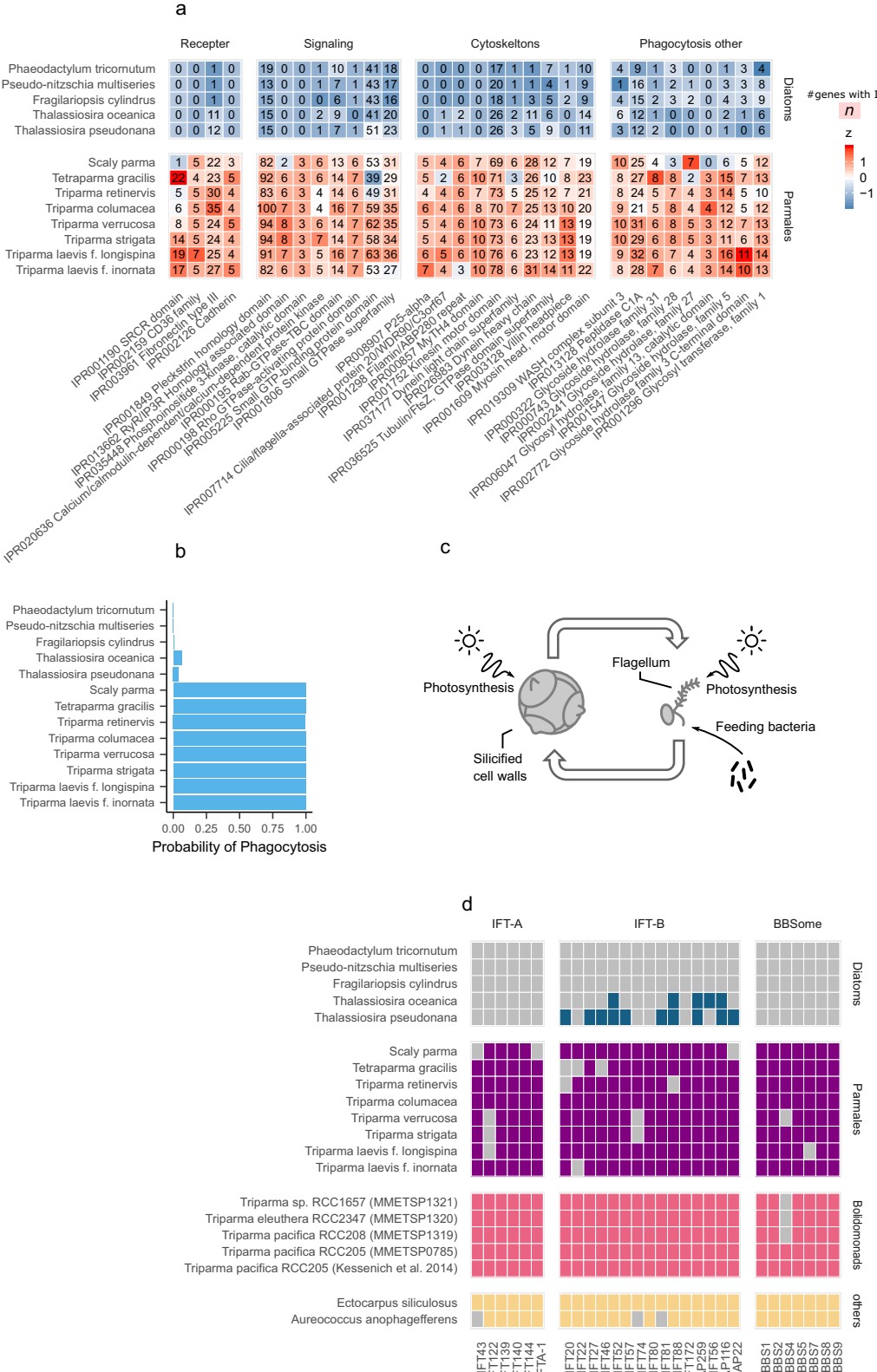

**Fig. 3 Phagotrophy and flagellum of parmaleans. a** InterPro domains enriched in diatom and parmalean genomes thought to be related to phagocytosis. The colours are scaled in ascending order from blue to red by the z-value in each row. Source data are provided as Supplementary Data 3. **b** Probability of phagotrophy predicted using a genome-scale tool developed by the Burns et al. (2019). Source data are provided as Supplementary Data 6. **c** Schematic view of hypothesized parmalean life cycle. **d** Presence (filled square) or absence (or loss: grey square) of genes/transcripts related to intraflagellar transport (IFT) subunits. Accessions can be found in Supplementary Data 7.

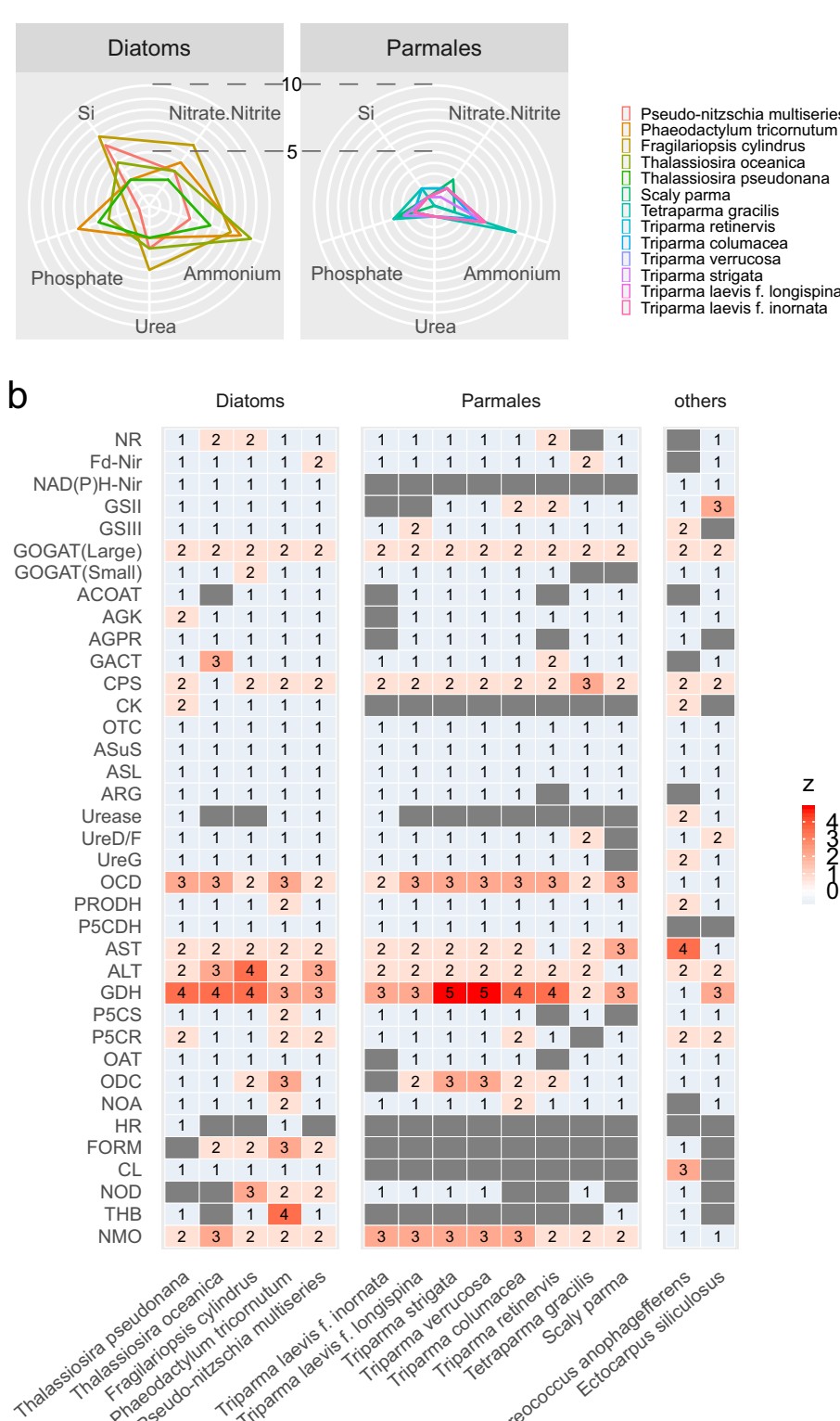

**Fig. 4 Ecophysiology of diatoms and parmaleans. a** Distribution of nutrient transporter genes. Each axis represents the number of nitrate/nitrite transporter, ammonium transporter, urea transporter, phosphate transporter, or silicic acid (Si) transporter genes. Accessions can be found in Supplementary Data 8. **b** Genes involved in nitrogen assimilation (including ornithine–urea cycle). The colours are scaled in ascending order from blue to red by the z-value in each row; a grey square indicates absence of the gene. Gene names are abbreviated; full names and accessions can be found in Supplementary Data 9.

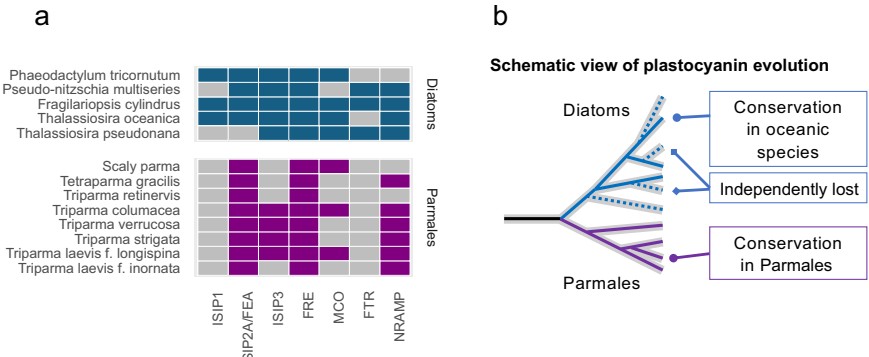

**Fig. 5 Iron-related genes of diatoms and parmaleans. a** Presence (filled square) or absence/loss (grey square) of iron uptake system genes. Gene names are abbreviated; full names and accessions can be found in Supplementary Data 10. **b** Schematic view of the evolutionary pattern of plastocyanin genes. A whole phylogenetic tree is shown in Supplementary Fig. 5.

adaptive strategy to differentially use cytochrome $c_6$ and plastocyanin. Phylogenetic analysis revealed that the plastocyanin genes from diatoms and parmaleans were monophyletic (with dictyochophytes and others), except for one from *Fragilariopsis kerguelensis*, which was grouped with bacteria (Supplementary Fig. 5). This result contradicts the previously proposed horizontal acquisition of plastocyanin genes in pelagic diatoms[52]. The diatom/ Parmales common ancestor likely possessed both cytochrome $c_6$ and plastocyanin, and some diatoms (mostly coastal ones) lost their plastocyanin (Fig. 5b).

**Silicate metabolism**. Each parmalean genome contained 1-2 silicic acid transporter (SIT) gene, whereas diatom genomes contained 3-8 SIT genes (Fig. 4a). Most SIT genes of diatoms encoded a 10-fold transmembrane type (i.e., single SIT domain), whereas many SIT genes of parmaleans encoded a 20-fold transmembrane type (i.e., two SIT domains). Phylogenetic analysis of SIT domains indicated that parmalean SIT genes belong to the most basal clade of diatom SITs (clade B)[54], and that the 20-fold transmembrane-type SITs of Parmales are the result of multiple domain duplications in the *Triparma* lineage (Fig. 6). A large number of paralogous SITs (at least five clades) in diatoms was generated through multiple gene duplications in the diatom lineage after it diverged from the Parmales lineage.

We also found silicanin homologs, some of which are biosilica-associated proteins[55], in parmalean genomes. The parmalean genomes contained low numbers (between 0 and 2 per species) of silicanin homologs, compared to between 4 and 13 in diatoms. Parmalean silcanin homologs have the RXL domain, which is typical of many diatom biosilica-associated proteins[56–59] but lack the NQ-rich domain that is found in the Sin1 and Sin2 genes of *Thalassiosira pseudonana*[55]. Silicanin homologs have been reported in transcriptome data of other non-diatom eukaryotes such as the ciliate *Tiarina fusus* and the dictyochophyte *Rhizochromulina marina*[55]. We also found 19 silicanin homologs from non-diatom eukaryote transcriptomes in the MMETSP database (15 sequences from *Tiarina fusus*, 2 from *Rhizochromulina marina*, 1 from the dinoflagellate *Durinskia baltica* and 1 from the dinoflagellate *Kryptoperidinium foliaceum*; *Durinskia* and *Kryptoperidinium* are known to have endosymbiont originated from diatoms[60], but their endosymbiont does not have silicified cell walls). Our finding of silicanin homologs in most of the analyzed parmaleans strongly suggests that the silicanin gene was already present in the diatom/Parmales common ancestor. Silicanins, like SITs, have undergone multiple gene duplications within the diatom lineage after the diatom/ Parmales divergence. Interestingly, SIT and silicanin proteins

were not found in any bolidomonad transcriptomes, which is consistent with their lack of silica plates.

**Ecological strategies and evolutionary scenarios**. By comparing the genomes of eight parmaleans and five diatoms, we were able to delineate differences and similarities in gene content between these two taxa (Fig. 7). Based on the gene-based trophic model, our analysis suggests that parmaleans are phago-mixotrophs that can acquire nutritional resources such as carbon, nitrogen, phosphorus, vitamins, and trace elements (e.g., iron) in the form of organic compounds by grazing other organisms, such as bacteria. Therefore phago-mixotrophs is considered less dependent on the uptake of inorganic nutrients than photoautotrophs. However this advantage is traded off with an associated increase in metabolic costs for incorporating and maintaining the cellular components required for both autotrophy and phagotrophy. In addition, since phagotrophy reduces the cell surface area for transporter sites, phago-mixotrophs are thought to have lower growth efficiency relative to photoautotrophic specialists[61,62]. According to a theoretical study, mixotrophy is beneficial especially in oligotrophic water, whereas autotrophy is advantageous in eutrophic environments[63,64].

Previous studies suggested that some mixotrophs can widen their niche by alternating their trophic strategies[65,66]. For example, several coccolithophores (Haptophyta) are known to alternate between a motile phago-mixotrophic haploid stage and a non-motile autotrophic diploid stage based on nutrient condition[67]. Based on these facts and other field data, it has been previously hypothesized that parmaleans have a similar life stage alternation[3]. Namely, parmaleans may live as silicified photoautotrophs during winter (the cold mixing season) when nutrients are rich, while they may feed on bacteria through phagocytosis as naked flagellates during summer (the warm stratified season) when nutrients are depleted. Our study reinforces the possibility of such a life cycle in Parmales, by detecting the genes for phagocytosis which has a potential association with the naked-flagellate stage. This putative life cycle may also explain the wide distribution range of some parmalean groups, from coastal regions to tropical, Arctic and Antarctic regions[2,3].

In addition to the absence of phagotrophy in diatoms, our analysis revealed a marked contrast in the gene repertoires between diatoms and parmaleans, with all indicating the autotrophic adaptations of diatoms. For example, there is a large difference in the number of nutrient transporter genes between diatoms and parmaleans (Fig. 4a), clearly representing an adaptation of diatoms to eutrophic environments, although it is

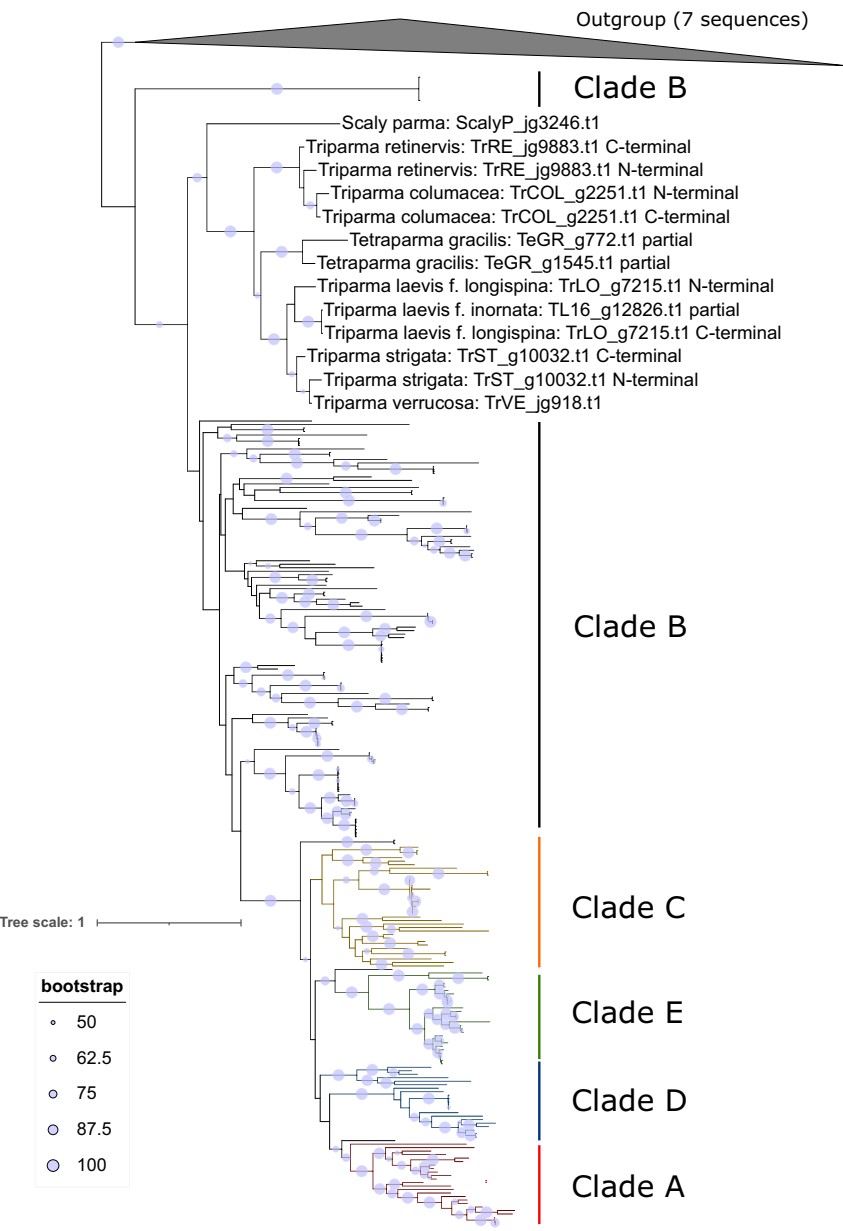

**Fig. 6 Phylogenetic tree of SIT domains.** Maximum likelihood phylogenetic tree of the SIT domains of diatoms, parmaleans, and ochrophytes (outgroup). Sequences with more than two SIT domains were separated to each domain and aligned. Grouping of paralogues from diatoms is based on the classification of Durkin et al.[55] Bootstrap values >50 are shown as circles on the branches. The parmaleans clade has been manually expanded to permit legibility.

not clear whether these paralogous genes have different functions (e.g., affinity, transport rates, and subcellular localization) or a dosage effect[68]. In addition, there are differences in the number of genes involved in biophysical carbon-concentrating mechanisms (Supplementary Fig. 6, Supplementary Data 11, Supplementary Data 12; See Supplementary Note). Diatoms possess higher $CO_2$ fixation capacity relative to other phytoplankton groups[69], and these gene repertoires may support this trait. We also revealed the expansion of protease and sulfotransferase genes in diatoms in addition to the previously described expansion of cyclin and heat-shock transcription factor genes[16,17] (Fig. 2a). These genes are likely involved in stress response and population control, which support the extraordinary growth capacity of diatoms.

To the best of our knowledge, phagotrophic mixotrophy has not been observed in diatoms, although osmotrophic mixotrophy is known in diatoms (e.g., *Phaeodactylum tricornutum* feeds on various carbon sources[70] and non-photosynthetic osmotrophs such as *Nitzschia putrida* also exist[71]). All diatoms that we studied were predicted as photoautotrophs (Fig. 3a, b) and other diatoms including *Epithemia pelagica* with endosymbiotic cyanobacteria[72] were predicted as the same from their genomic data (see Supplementary Note). On the other hand, diatoms have secondary plastids originated from red algae, suggesting phagotrophy must have existed for the ancestor of diatoms to take up them. Some members of ochrophytes, such as chrysophytes and dictyochophytes[73], are known to be phago-mixotrophs and our results suggest that Parmales, which is the closest group to diatoms, is also phago-mixotrophs. These facts firmly support the ideas that diatom/Parmales common ancestor was phago-mixotrophs, and there were massive loss of phagocytosis-related genes and specialization to photoautotrophy in the early evolution of diatoms after diverging from

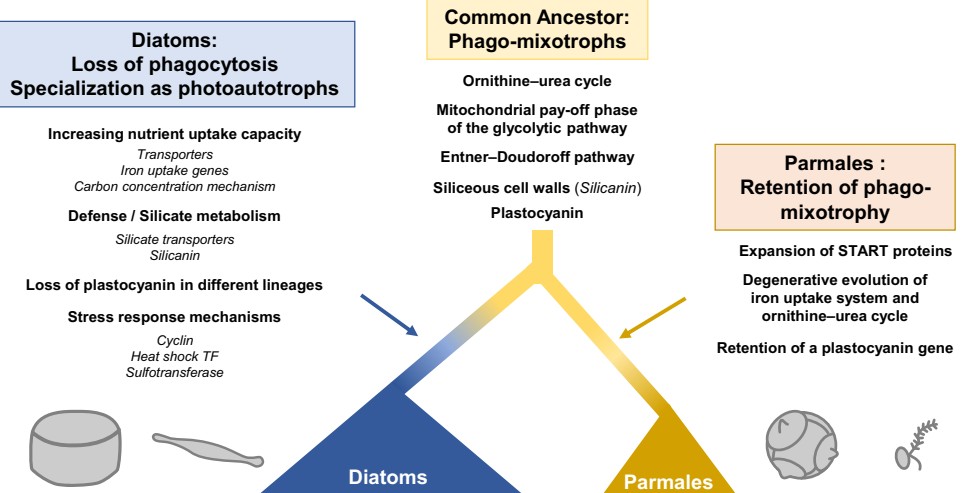

**Fig. 7 Schematic view of diatoms and Parmales evolution.** Putative evolutionary history of diatoms and Parmales identifying components contributing to their ecophysiology.

Parmales (ca. 200 million years ago[7]) but before the subsequent diversification of diatom lineages (i.e., the crown age; ca. 190 million years ago[7]).

Diatoms always have silicified cell walls in the vegetative stage, whereas parmaleans putatively switch between two life stages, silicified/non-flagellated and naked/flagellated stages. The silicified cell wall could provide a barrier against grazers, parasites, and pathogens[74], but is obviously incompatible with phagocytosis as it completely covers the cell. Thus, there is a trade-off between silicification/autotrophy and phagocytosis, and the loss of phagotrophy in diatoms may have been related to benefits from the silicified cell wall. To reveal why photoautotrophic diatoms diverged from the phago-mixotrophic lineage and specialized to the silicified life stage, it is necessary to understand not only the costs and benefits associated with mixotrophy but also those of defence by silicified cell walls.

The next possible step in the evolution of diatoms after specialization to silicification and photoautotrophy might have been to thicken their silicified cell wall and increase their cell size[9]. Diatoms tend to have larger cell sizes than parmaleans, and the evolution of these traits has the great advantage of increasing resistance to grazers[75]. The evolution of silicic acid transporter genes (Fig. 6) may have supported the evolution of silicified cell walls because diatoms with thick walls and large cells require large amounts of silicate. It is also known that nutrient metabolism, especially nitrogen metabolism, is closely related to silica deposition in diatoms[76]. Thus, the ability of diatoms to take up nutrients may also be related to the evolution of their silicified cell wall. Silicanin, which diversified in diatoms, is also known to be related to the strength and stiffness of their cell walls[77] and may have been important in the precise control of the formation of thick cell walls. It has been also pointed out that vacuoles play a major role in cell size expansion[9]. However, there is little evidence of differences in vacuole-related genes between parmaleans and diatoms (e.g., lack of a vacuolar nitrate transporter ortholog in parmaleans), so further discovery and analysis of the relevant genes are needed to address this issue.

Diatoms also have important systemic impacts on marine iron usage, often dominating iron-stimulated blooms[78]. Analyses of iron utilization strategies revealed that the ISIP1 gene, which is involved in siderophore-mediated iron acquisition, is absent in parmaleans and specific to diatoms (Fig. 5a). Siderophores are thought to be major components of microbial iron cycling in the ocean[79]. The lack of the ISIP1 gene in parmaleans supports the idea

that this gene underlies the high iron uptake capacity of diatoms and supports their photoautotrophic lifestyle. We also found that plastocyanin, which is an alternative for iron-requiring proteins in photosynthesis, is widely distributed in parmaleans. Phylogenetic analysis suggests that each lineage of diatoms lost their plastocyanin genes independently, and that pelagic diatoms and parmalean groups conserved plastocyanin genes from their common ancestor (Fig. 5b). This, together with their life cycles, may explain the wide distribution range of some parmalean group, including coastal and Arctic regions as well as iron-deficient areas such as tropical open ocean and Antarctic regions. Parmaleans retained plastocyanin to balance their restricted capacity for iron uptake in iron-limited environments; diatoms increased their iron uptake capacity (e.g., ISIP1), while several lineages have specialized to coastal eutrophic environments and lost plastocyanin.

Our analysis also revealed that the ornithine–urea cycle, the mitochondrial pay-off phase of the glycolytic pathway, and the Entner–Doudoroff pathway, which have been cited as unique features of diatoms, were substantially conserved from the common ancestor of Parmales and diatoms (Fig. 4b, Fig. 7, Supplementary Fig. 3, Supplementary Fig. 7, Supplementary Data 9, Supplementary Data 13: see Supplementary Note). We also found the expansion of genes related to lipid metabolism and intracellular signalling, and the degenerative evolution of several genes related to iron uptake and ornithine–urea metabolism in Parmales (see Supplementary Note). However, their physiological functions and evolutionary significances remain unclear. Future studies based on a larger set of genomic data will further enhance understanding of the physiology, ecology, and evolution of these fascinating organisms.

## Methods

**Culture**. We used strains of the parmaleans *Triparma laevis* f. *inornata* (NIES-2656; Microbial Culture Collection at the National Institute for Environmental Studies, Japan), *Triparma laevis* f. *longispina* (NIES-3699), *Triparma verrucosa* (NIES-3700), and *Triparma strigata* (NIES-3701), isolated from the Oyashio region of the western North Pacific. For the other strains, water samples were collected at 10 m in the Notoro-ko lagoon (44°3'2.1" N, 144°9'38.8" E, December 2015) for *Triparma retinervis*, at 10 m in the Sea of Okhotsk (45°25'0" N, 145°10'0" E, June 2017) for *Tetraparma gracilis* and *Triparma columacea*, and at 30 m in the Sea of Okhotsk (44°30'0" N, 144°20'0" E, June 2014) for the uncharacterized 'Scaly parma'. The strains were isolated by serial dilution with siliceous cell wall labelling techniques described previously[5]. The strains were cultured in f/2 medium[80] at 5 °C under a light intensity of ca. 30 μmol photons $m^{-2}$ $s^{-1}$ (14:10 L:D cycle).

**Genomic DNA, RNA extraction and sequencing**. Cells grown under exponential growth phase were harvested by centrifugation, and either DNA (all strains, except

*Triparma laevis* f. *inornata*) or RNA (for *Triparma laevis* f. *inornata*, *Triparma verrucosa*, *Triparma retinervis* and 'Scaly parma') was extracted using the DNeasy Plant Mini Kit or RNeasy Plant Mini Kit (Qiagen, Venlo, Netherlands), respectively. Libraries were generated using the Illumina TruSeq DNA/RNA sample preparation kit (Illumina, Inc., San Diego, USA). Sequencing of whole genomes or transcriptomes was performed on an Illumina HiSeq X (150 bp, paired-end) or HiSeq 2000 (100 bp, paired-end), respectively. Exceptionally, the genome of *Triparma laevis* f. *longispina* and 'Scaly parma' was sequenced with an Illumina HiSeq 2500 (150 bp, paired-end). DNA extraction and sequencing methods for *Triparma laevis* f. *inornata* were already reported in Kuwata et al.[14]

**Genome assembly and microbial sequence contamination removal**. Genome assembly and contamination removal methods for *Triparma laevis* f. *inornata* were already reported in Kuwata et al.[14] For the other strains, the Illumina reads were trimmed with Trimmomatic (v.0.38)[81] using the following parameters: LEADING:20 TRAILING:20 SLIDINGWINDOW:4:15 MINLEN:36 TOPHRED33. The filtered reads were assembled by Platanus (v.1.2.4)[82] with default options. To remove bacterial contamination from contigs, clustering of contigs was performed based on the coverage calculated by read mapping, GC frequency, and k-mer frequency. In addition, the phylogenetic classification of the genes in the contigs was estimated using lowest common ancestor analysis. The results were used to determine the clusters composed of bacterial contigs. Read mapping to assembled contigs with the filtered reads was performed with BWA (v.0.7.17)[83]. The coverage was calculated from the resulting.sam file using sam_len_cov_gc_insert.pl (https://github.com/sujaikumar/assemblage), which was also used to determine the GC content. The tetramer frequency of contigs was calculated using cgat (v.0.2.6)[84]. Open reading frames (ORFs) were predicted using GeneMarkS (v.4.30)[85] and their taxonomy was annotated with a last common ancestor strategy as in Carradec et al.[86] ORFs were searched against a database composed of UniRef 90[87], MMETSP database[88], and Virus-Host DB[89] using DIAMOND (v.0.9.18)[90]. Selected hits were then used to derive the last common ancestor of the query ORFs with the NCBI taxonomy database. Clustering of contigs was performed using the R script provided in the CoMet workflow[91] with coverage, GC content, and k-mer frequency as information sources. The organism from which each cluster originated was determined from the estimated phylogeny of the genes in the contigs belonging to the cluster. Contigs belonging to bacterial-derived clusters were excluded from the datasets and not used in downstream analyses.

We also performed a BLASTn (v.2.11.0) search against the organelle genomes of *Triparma laevis* f. *inornata*[92] to remove the organelle genome from assembled contigs. Contigs that hit the organelle genome of *Triparma laevis* f. *inornata* with *E*-values < 1e − 40 were excluded from our dataset as organelle genomes.

**Genome annotations**. For *Triparma laevis* f. *inornata* genome[14], rRNA and tRNA genes were predicted by Barrnap (v.0.6, http://www.vicbioinformatics.com/software.barrnap.shtml) and tRNA-scan-SE (v.1.23)[93], respectively. The protein coding-genes were predicted by AUGUSTUS (v.3.2.2)[94] with the RNA-seq data mentioned above. First, the RNA-seq reads processed by fastx-toolkit (v.0.0.13, http://hannonlab.cshl.edu/fastx_toolkit/) were mapped to the contig of the *Triparma laevis* f. *inornata* nuclear genome and assembled into transcript contigs using Tophat (v.2.1.1)[95], Cufflinks (v.2.2.1)[96] and Trinity (v.2.0.6)[97], respectively. The diatom protein sequences from *Thalassiosira pseudonana*[16] and *Phaeodactylum tricornutum*[17] were subsequently aligned to the transcript contigs using tBLASTn search (v.2.2.29) and Exonerate (v.2.4.0)[98] for detecting CDS regions in the *Triparma laevis* f. *inornata* genome. Finally, a total of 687 loci on the *Triparma laevis* f. *inornata* contigs were selected as those carrying full-length CDSs and used for parameter fitting in training hidden Markov models in AUGUSTUS. In gene prediction, the mapping data from both RNA-seq reads and diatom protein sequences were utilized as hints in AUGUSTUS.

For other genomes, tRNA genes were predicted using tRNAscan-SE (v.2.0.7)[99]. Non-coding RNAs excluding tRNAs but including rRNAs were predicted with the Rfam database using infernal (v.1.1.3)[100]. Repeats and transposable elements were annotated and soft-masked using RepeatModeler (v.2.0.1)[101] and RepeatMasker (v.4.1.0)[102]. For *Triparma verrucosa*, *Triparma retinervis* and 'Scaly parma', the protein-coding genes were predicted by BRAKER2[103] with the RNA-seq data mentioned above and a reference protein sequence database. We generated a reference protein sequence database for BRAKER2 from OrthoDB[104], MMETSP database[88] and *Triparma laevis* f. *inornata* protein sequences predicted previously. Firstly, RNA-seq data were mapped to the contigs using STAR (v.2.7.3a)[105], generating a .bam file. Secondly, BRAKER2 was run in –etpmode with the generated .bam file and the reference protein sequence database as the protein hints. For *Triparma laevis* f. *longispina*, *Triparma strigata*, *Triparma columacea* and *Tetraparma gracilis*, the protein-coding genes were predicted by BRAKER2 only with a reference protein sequence database. We updated the mentioned reference protein sequence database with the predicted protein sequences form *Triparma verrucosa*, *Triparma retinervis*, and 'Scaly parma', and generated a new database. Finally, BRAKER2 was run in -epmode using the newly generated reference protein sequence database as the protein hints.

The completeness of genome assemblies and gene predictions were evaluated using BUSCO (v5.1.2)[106] with the stramenopiles_odb10 dataset.

**Functional annotation**. For methodological consistency, we applied the same annotation pipelines for our novel genomes and the genomes downloaded from public databases. For each genome, we used CD-HIT (v.4.8.1)[107] with the parameters -c 1 -aS 1 to remove protein sequences with 100% similarity for downstream analysis. Genes were functionally annotated by InterProScan (v.5.26-65.0)[108] and eggNOG-Mapper (v.2.0.1)[109] with the eggNOG database (v.5.0)[110]. Protein localization was predicted using MitoFates (v.1.1)[111], TargetP (v.2.0)[112], SignalP (v.4.1)[113], and ASAFIND (v.1.1.7)[114]. Protein functions and localizations were manually curated for detailed analyses.

**Phylogenetic analysis**. For phylogenetic analysis using 18 S rRNA, we downloaded 18 S rRNA genes categorized as "Bolidomonas" from the SILVA database (accessed May 2020)[115]. For 'Scaly parma', *Triparma columacea* and *Triparma retinervis*, which are missing from downloaded dataset, we assembled 18 S rRNA gene from raw DNA sequences data using PhyloFlash (v 3.4)[116]. We merged these two datasets and removed shorter sequences than 900 bp and add some diatoms sequences as outgroups. We aligned and masked the sequences using SSU-ALIGN (v 0.1.1)[117] with default parameters. A maximum likelihood tree was inferred with the generated multiple sequence alignment by IQ-Tree2(v 2.2.0)[118] with the GTR + I + G model. We performed 1,000 ultrafast bootstrap replicates.

For phylogenomic analysis, orthologous genes (OGs) were determined by OrthoFinder (v.2.3.7)[119] with protein sequences of 8 parmalean genomes, other available 10 stramenopile genomes, and ochrophyte transcriptomes (Supplementary Data 14) from the MMETSP database[88] and Kessenich et al.[120] Gene annotation was not available for the data from Kessenich et al.[120] therefore, coding sequences were annotated using TransDecoder (v.5.5.0) (https://github.com/TransDecoder/TransDecoder). Only single-copy genes in each OG and genes that were found in the 18 stramenopile genomes were retained for downstream phylogenomic analysis, resulting in 175 OGs. Gene sequences within each OG were aligned using MAFFT (v.7.453)[121] in the linsi mode, and poorly aligned regions from the multiple sequence alignment were removed by trimAl (v.1.4.1)[122] in the automated1 mode. The resulting supermatrix contained 55,777 amino acid positions for 18 species, with 6.09 % missing data. A maximum likelihood tree was inferred by RAxML (v.8.2.12)[123] with the partition information of each gene and the LG + F model. We performed 1,000 bootstrap replicates and all bootstrap values were 100, indicating full support.

**Predictions of phago-mixotrophy using a gene-based model**. Predicted protein data from eight parmalean genomes and five diatom genomes were tested for phagocytotic potential using a gene-based model described by Burns et al.[28] To determine the phagocytotic potential of parmaleans, we also tested the five transcriptomes of the naked flagellate (bolidomonads) from the MMETSP database[88] and Kessenich et al.[120]

**Phylogenetic analysis of silicon transporter domains**. We used the sequence data of SIT proteins from diatoms and ochrophytes provided by Durkin et al.[54] in addition to those of parmaleans determined in this study. Diatom and parmalean SIT proteins are usually composed of a single SIT domain, but some contain more than two domains. To analyse multiple domains at once, SIT domain regions were determined using hmmscan (HMMER.3.3.2)[124] with PF03842 from Pfam using the profile's GA gathering cutoff (--cut_ga mode) and selected for downstream analysis. Each SIT domain sequence was aligned using MAFFT (v.7.453)[121] in the linsi mode with default parameters and unreliable sequences were manually removed. A maximum likelihood phylogenetic tree was inferred from this multiple alignment using RAxML (v.8.2.12)[123] with default parameters. The amino acid substitution model was automatically determined to be the LG model by the software. Bootstrap values were obtained based on 100 bootstrap replicates.

**Identification of silicanin homologs**. To find silicanin homologs in MMETSP database[88], and our genomes, we used BLASTp search (Blast+ v2.10.1) with Sin1 gene of *Thalassiosira pseudonana* as query and default parameters. Among the hit sequences, we selected those with *E*-value < 1e-5 and >300 aa and finally obtained 1990 silicanin homologs.

**Phylogenetic analysis of plastocyanin**. We used hmmsearch (HMMER v.3.3)[124] with TIGR02656.1 from TIGERFAMs using the profile's GA gathering cutoff (--cut_ga mode) to find plastocyanin genes in Uniref 90[87], MMETSP database[88], and our genomes. The genes from MMETSP database clustered with 97% similarity using CD-HIT (v.4.8.1)[107]. Unreliable sequences were removed manually. We next obtained 716 plastocyanin genes of photosynthetic eukaryotes, cyanobacteria, and cyanophages. Because of the large divergence of the sequences and small number of alignable regions, we used gs2, a software to conduct the Graph Splitting (GS) method[125], which can resolve the early evolution of protein families using a graph-based approach, to estimate the phylogenetic tree of plastocyanin. We ran the GS method with 100 replicates using the Edge Perturbation method for statistically evaluating branch reliability.

**Statistics and reproducibility**. Significant differences in protein domain content annotated by InterProScan between the compared genomes were identified using Fisher's exact test (two-sided) to calculate the *p-value* for the difference in the number of genes with each InterPro domains between parmalean ($n = 8$) and diatom genomes ($n = 5$). The *p-values* were corrected for multiple comparisons using Bonferroni correction. Then, we manually selected and grouped the domains that are involved in specific biological processes.

Most analyses in this study have been performed using R (v.3.6.1)[126]. All other programs used in this study are provided in the Methods section.

**Reporting summary**. Further information on research design is available in the Nature Portfolio Reporting Summary linked to this article.

## Data availability

In this study we used several public genomic data of diatoms[16,17,127,128] and other stramenopiles[129–133], as well as transcriptome data[88,120] (Supplementary Data 15). Sequence data generated during the current study are available in DDBJ bioprojects, under accession number PRJDB14101 (RNA reads for *Triparma laevis* f. *inornata*), PRJDB13844 (DNA reads for the other seven strains), and PRJDB13933 (RNA reads for the other three strains). The assembly data analysed during the current study are also available in the DDBJ repository, under accession numbers BLQM01000001-BLQM01000902 (*Triparma laevis* f. *inornata*), BRXW01000001-BRXW01001055 (*Triparma laevis* f. *longispina*), BRXX01000001-BRXX01000659 (*Triparma verrucosa*), BRXY01000001-BRXY01000634 (*Triparma strigata*), BRXZ01000001-BRXZ01008760 (*Triparma retinervis*), BRYA01000001-BRYA01001858 (*Triparma columacea*), BRYB01000001-BRYB01007082 (*Tetraparma gracilis*), and BRYC01000001-BRYC01001921 ('Scaly parma'). The data underlying our findings and numerical source data for graphs and charts are provided in Supplementary Data 1–15. The newly generated 18 S rRNA gene sequences of Parmales from this study are available on our website (https://www.genome.jp/ftp/db/community/parmales_diatoms/). All other data are available from the corresponding author on reasonable request.

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

## Acknowledgements

This work was supported by JSPS/KAKENHI (No. 22657027, 23370046, 26291085, 221S0002, 16K07489, 16H06279 (PAGS), 17H03724), the Canon Foundation, the Collaborative Research Program of Institute for Chemical Research, Kyoto University (No. 2016-30, 2015-39), and the JST "Establishment of University Fellowships Towards The Creation of Science Technology Innovation" Grant Number JPMJFS2123. Computational time was provided by the SuperComputer System, Institute for Chemical Research, Kyoto University. We thank Gabe Yedid, Ph.D., from Edanz (https://jp.edanz.com/english-editing-b) for editing a draft of this manuscript. We thank Drs. Adriana Lopes dos Santos and Daniel Vaulot for valuable suggestions.

## Author contributions

H.B. performed most of the bioinformatics analyses presented in this work and wrote initial version of the manuscript. R.B.-M., H.E., and H.O. supervised the bioinformatics part of the study. Y.N. performed genome assembly and gene prediction for *Triparma laevis* f. *inornata*. A.K. coordinated the genome sequencing part of the study. S.S., S.Y., K.Y., M.I., and A.K. contributed to culture and DNA/RNA sequencing. N.S. contributed to functional interpretation of the genomes. All authors contributed to the interpretation of the results and the finalization of the manuscript.

## Competing interests

The authors declare no competing interests.
