## [Peer Review File · Communications Biology]

Reviewers' comments:

Reviewer #1 (Remarks to the Author):

This is an interesting study, providing a significant advance into our understanding of the deep evolution of diatoms, alongside the ecological significance of Parmales in the contemporary ocean. I nonetheless have some suggestions for additional analyses and/ or existing elements of the paper that could be better emphasised, to give more clarity to the presented conclusions.

Evolutionary context:

- It would be helpful to redraw the concatenated phylogeny with the presence of *Bolidomonas* transcriptomes from MMETSP. Even if these are not used for subsequent gene gain/ loss questions, it would be useful to verify that the genomes presented cover a significant proportion of Parmalean diversity rather than a single derived group.
- It would likewise be worth using a phylogenetically-aware program (e.g., phyl-PCA or CAFE) to verify to what extent the gene family gains/ losses/ expansions/ contractions can be attributed to the last common ancestor of diatoms, or of Parmales, as opposed to more recent derivations in individual species

Environmental significance:

- Have the authors looked for the sequenced species in TARA Oceans, and if so where are they found? If they are restricted to Fe-rich regions and/ or show low occurrence in the Southern and South Pacific oceans this may provide some context to the absence of Fe-dependent metabolism pathways
- There are also some MAGs from TARA Oceans that resolve with bolidomonads: what do these show in terms of genome content compared to the isolated species?

Functional biology:

- I note that the authors have performed *in silico* localisation predictions for much of the discussed proteins. It would be helpful to emphasise this more in the text, particularly in the context of the secretory and endocytic machinery, which may provide context into the loss of phagotrophy in diatoms, or the alternating haplodiploid lifestyle in Parmales.
- N metabolism: I would welcome some discussion for the presence (or not) of a C4 shunt in Parmales, given its implication in Asp shuttling from the diatom PPC to mitochondria (c.f., Gu et al. *New Phytol* 2022). I would likewise like to see some discussion of the predicted localisations (plastid, mitochondria, or two discrete paralogous pathways) of Asp, Ala and Glu aminotransferases, given their likely duality and importance in N recycling in diatoms (c.f. Smith et al., *N Comms* 2019)
- Fe metabolism: A minor point, but NB that while ISIP1 in the classical sense is principally associated with diatoms, Kazamia et al. 2018 document probable distant homologues in pelagophytes, haptophytes and Kareniacean dinoflagellates.

Reviewer #2 (Remarks to the Author):

The manuscript by Ban et al. focuses on a group of eukaryotic marine algae known as Parmales. These organisms alternate between a life cycle stage in which each cell is surrounded by silica plates assumed to provide protection against grazing, and a flagellated stage where cells are assumed to have the capacity to carry out mixotrophy and consume bacteria. This lifestyle stands in contrast to the ubiquitous diatoms, a sister group that has lost the ability to consume bacteria and is characterized by cell walls of silica. The authors used genome comparisons of 5 model diatoms with 8 sequenced Parmales genomes to explore the evolutionary basis of the major lifestyle differences between these two groups of diatoms. The authors propose a tradeoff between retaining mixotrophy (flagellated cells of the Parmales) and building grazer defenses (silica cell walls of diatoms). This is an important conclusion as mixotrophy is a common feature of marine ecosystems and yet diatoms, one of the more successful phytoplankton, have lost mixotrophic capabilities. The manuscript is well-written with the key points clearly laid out. I greatly enjoyed reading it and learned a lot.

I have a few minor suggestions outlined below to clarify points.

1. Line 101. You should clarify that the 164 orthologous genes used for your analysis were identified in your study not based on previous work. I only realized this after reading the methods section.
2. Figure 2. Initially, I was not sure what the numbers in Fig. 2a and 2d referred to. I assume they indicate the number of InterPro domains in category in each organism. If correct, this should be made explicit in the legend.

The following statement in the 2a legend should be clarified. "We manually clustered and selected the domains that appear to be involved in a specific process." I think the authors could simply include (see methods).

Legend for 2d includes the statement "InterPro domains that were not significant but were considered important are marked with an asterisk (*)." The domain that is not significantly enriched in the Parmales genomes should be removed from the figure.

3. Figure 3. Fig 3a. An axis scale for this figure should be added. Fig. 3b legend. Need to indicate where full gene names can be found (as was done for 3c).
4. Figure 4 legend. The authors state "only important bootstrap values are noted." Instead, the authors should state bootstrap values greater than X are shown, with X defined by the authors.
5. The authors should make available their "custom perl scripts"
6. Please add the parameters used for the HMM, MAFFT, and RAXML.
7. Please spell out the full name of the culture center.
8. Line 206-209 could benefit from a supplemental schematic.

Reviewer #3 (Remarks to the Author):

Ban et al. generated seven parmalean genome assemblies and performed comparative genomic analysis between parmales and diatoms to dissect the genetic basis of adaptation in these two groups of organisms. The manuscript is straight forward and well-written. It has interesting and important results for the readers of Communications Biology. However, I have some minor comments to the manuscript.

Line 111. Not having InterPro domains does not indicate function of a protein is unknown. If the authors want to conclude that the lineage-specific genes of diatoms and parmaleans are functionally unknown, they should search the proteins in more datasets.

Line 131. It would be better that the authors explain more why the enrichment of intracellular signaling pathways associate putative alternating life cycle stages in parmaleans.

Line 268. It is insufficient to conclude that silicanin genes in ciliate and dinoflagellate are derived from diatoms through HGT without phylogenetic analysis.

Referee #1: protist cell biology and evolution

Referee #2: phytoplankton eco-physiology

Referee #3: genomics, marine biology

Reviewers' comments:

Reviewer #1 (Remarks to the Author):

This is an interesting study, providing a significant advance into our understanding of the deep evolution of diatoms, alongside the ecological significance of Parmales in the contemporary ocean. I nonetheless have some suggestions for additional analyses and/ or existing elements of the paper that could be better emphasised, to give more clarity to the presented conclusions.

Thank you for your comments. We are glad to know that you recognize the importance of our work. We answer to your specific comments below.

Evolutionary context:

- It would be helpful to redraw the concatenated phylogeny with the presence of Bolidomonas transcriptomes from MMETSP. Even if these are not used for subsequent gene gain/ loss questions, it would be useful to verify that the genomes presented cover a significant proportion of Parmalean diversity rather than a single derived group.

Thank you for your valuable suggestion. We newly reconstruct a phylogenetic tree of 18S rRNA genes including isolated bolidomonas and other related environmental sequences (Supplemental Fig. 1). This analysis clearly shows that our genomic data cover the wide diversity of Parmales including the most basal clade, albeit with a few minor exceptions such as the Env clade IIIa which includes two isolated bolidomonas algae.

Supplemental Fig. 1 is follows:

- It would likewise be worth using a phylogenetically-aware program (e.g., phyl-PCA or CAFE) to verify to what extent the gene family gains/ losses/ expansions/ contractions can be attributed to the last common ancestor of diatoms, or of Parmales, as opposed to more recent derivations in individual species

Thank you for your suggestion. We agree that distinguishing deep evolutionary events from more recent ones is an important point as you suggest, since our study is focusing on the characterization of the former events (deep evolutionary events that happened right after the separation of diatom and parmalean lineages). For this purpose, we took an approach to compare properties conserved in diatom genomes with those conserved in

parmalean genomes. We believe that our approach is appropriate for our purpose and robust enough to detect such deep evolutionary events, given the wide phylogenetic coverage of our genome samples. We agree that investigating losses/gains of genes along the entire phylogeny from the root to the leaves would be an important next step to characterize the evolution of these organisms, but we consider it is out of the scope of our present work. For this reason, we decided not to follow your suggestion. We hope you could agree with our standpoint.

Environmental significance:

- Have the authors looked for the sequenced species in TARA Oceans, and if so where are they found? If they are restricted to Fe-rich regions and/ or show low occurrence in the Southern and South Pacific oceans this may provide some context to the absence of Fe-dependent metabolism pathways

We appreciate your comment. We looked into again the work by Kuwata et al. 2018, which shows that the distribution of parmaleans is not restricted to Fe-rich regions, and some species show rather broad distributions (from coastal to HNLC regions). Kuwata et al. suggested that this distribution may be caused by intraspecific diversity. However, given the evidence of phagocytosis for all parmaleans (and bolidomonas), we hypothesize that this wide distribution is achieved by mixotrophy and an adaptive strategy to iron-deficient environments (e.g., plastocyanin). According to your comments, we added new sentences discussing this hypothesis in the manuscript (L.317-319, L.380-382).

L.317-196: “This putative life cycle may also explain the wide distribution range of some parmalean groups, from coastal regions to tropical, Arctic and Antarctic regions^{2,3}”

L.380-382: “This, together with their life cycles, may explain the wide distribution range of some parmalean group, including coastal and Arctic regions as well as iron-deficient areas such as tropical and Antarctic regions.”

- There are also some MAGs from TARA Oceans that resolve with bolidomonads: what do these show in terms of genome content compared to the isolated species?

We appreciate your comment. We newly analyzed two MAGs of bolidomonads determined from Tara Oceans and confirmed that these MAGs also have a phagocytosis signal and genes for silica metabolism. These results were added to the Supplemental

Text (L.18-L.32 of the Supplementary Text), which reads as follows.

“Comparison of the isolated genomes and Metagenome Assembled Genomes (MAGs)
To confirm the genomic features of paramleans identified in this study based isolate genomes, we examined two parmalean Metagenome Assembled Genomes (MAGs) from Tara Oceans Eukaryotic Genomes1 (TARA_ARC_108_MAG_00221, TARA_PON_109_MAG_00217). Both MAGs were predicted as phago-mixotrophs by a gene-based phago-mixotrophy prediction model (scores; 0.99 and 0.94 respectively). In addition, these MAGs had one silicic acid transporter (SIT) gene each (TARA_ARC_108_MAG_00221_000000002984.2.2, TARA_PON_109_MAG_00217_000000002599.19.1), as well as silicanin homologs (TARA_ARC_108_MAG_00221_000000001564.2.1, TARA_ARC_108_MAG_00221_000000002201.2.2, TARA_PON_109_MAG_00217_000000002307.2.1), which are missing from the bolidomonads (flagellate parmaleans) transcriptomes. These results indicate that the parmaleans living in nature represented by the MAGs undergo phagocytosis and produce silicified cell walls. This is consistent with the observations obtained from isolate genomes.”

Functional biology:

- I note that the authors have performed in silico localisation predictions for much of the discussed proteins. It would be helpful to emphasise this more in the text, particularly in the context of the secretory and endocytic machinery, which may provide context into the loss of phagotrophy in diatoms, or the alternating haplodiploid lifestyle in Parmales.

As you suggest, a systematic comparison of protein localization sites between diatoms and Parmales may provide a global view of architectural or functional differences between these two different types of cells. There could be such differences due to the presence/absence of specific genes (ex. phagocytosis-related genes). However, even if there are differences in the protein localization between the lineages, it would be difficult to attribute the differences to specific cellular traits. Furthermore, we detailed the functions and functional categories for phagocytosis and meiosis (Figure 2d and 2g). These functional descriptions provide some information about their localizations. We consider that providing a global picture of differences in predicted protein localizations can only provide a more ambiguous picture than the information provided in these figures

and texts. For this expected difficulty at the level of interpretation, and due to the fact that we have already described detailed functional differences for specific systems (phagocytosis/meiosis), we decided not to follow your suggestion. We hope you could understand our standpoint.

- N metabolism: I would welcome some discussion for the presence (or not) of a C4 shunt in Parmales, given its implication in Asp shuttling from the diatom PPC to mitochondria (c.f., Gu et al. *New Phytol* 2022). I would likewise like to see some discussion of the predicted localisations (plastid, mitochondria, or two discrete paralogous pathways) of Asp, Ala and Glu aminotransferases, given their likely duality and importance in N recycling in diatoms (c.f. Smith et al., *N Comms* 2019)

Thank you for your comment. The paper you suggested (we suppose Yu et al. *New Phytol* 2022) discussed C4 shunt in *P. tricornutum* by focusing on PEPC localized to mitochondria. It is an important topic to explore the evolutionary process of that pathway in diatoms/parmaleans. Interestingly, parmaleans also had two PEPC paralogs each, one of which was predicted to be targeted to mitochondria except ‘Scaly parma’ (Supplemental Fig. 6). However, it is difficult to prove the existence of the C4 pathway in parmaleans based on protein localization predictions alone (like in other previous studies, c.f., Kroth et al., *PLoS One*, 2008), and experimental evidence is essential (e.g., Yu et al. 2022). Furthermore, the localizations of genes involved in the C4 shunt (PEPC/MDH/PPDK) were not predicted to be the same within diatoms (Supplemental Fig. 6), indicating that results on *P. tricornutum* from Yu et al. may not be generalized across diverse diatoms and could not be attributed to their ecological success. Since we focused on delineating differences between diatoms and parmaleans and discussing this topic would be still highly speculative, we decide not to discuss this topic.

The evolution of nitrogen metabolism, especially intercellular partitioning, is also an important topic. Parmaleans have genes for the Urea cycle and aminotransferases (AST/ALT/GOGAT, sometimes in multiple copies) like diatoms. It is also interesting that most parmaleans have two ALTs, both with high similarity to the ALT1/ALT2 of *P. tricornutum* (grouped in the same EggNOG) (Supplemental Data. 6). However, like the situation discussed above on the C4 shunt, the predicted cellular localizations of aminotransferases are not consistent across diatoms (Supplemental Fig. 3), making it difficult to generalize the conclusions of Smith et al. 2019 to all diatoms. Since we are focused on delineating differences between diatoms and parmales, we decide not to

discuss this topic in this work. We hope you could understand our standpoint.

For this discussion, we carefully re-examined the presence of the ALT gene in each genome and revised Fig. 3b, Supplemental Fig. 3, and Supplemental Data.6.

Fig. 3b has been changed as follows.

Supplemental Fig. 3 has been changed as follows:

- Fe metabolism: A minor point, but NB that while ISIP1 in the classical sense is principally associated with diatoms, Kazamia et al. 2018 document probable distant homologues in pelagophytes, haptophytes and Kareniacean dinoflagellates.

Thank you for your comment. We changed the text (L.244-247) to clarify this point as follows.

“Our results support the idea that ISIP1 is a diatom-specific gene with some exceptions (possible homologs in some species of pelagophytes, haptophytes, and dinoflagellates within the genus *Karenia*)⁴⁸ and its presence may underlie diatoms’ high iron uptake capacity.”

Reviewer #2 (Remarks to the Author):

The manuscript by Ban et al. focuses on a group of eukaryotic marine algae known as Parmaeles. These organisms alternate between a life cycle stage in which each cell is surrounded by silica plates assumed to provide protection against grazing, and a flagellated stage where cells are assumed to have the capacity to carry out mixotrophy and consume bacteria. This lifestyle stands in contrast to the ubiquitous diatoms, a sister group that has lost the ability to consume bacteria and is characterized by cell walls of silica. The authors used genome comparisons of 5 model diatoms with 8 sequenced Parmales genomes to explore the evolutionary basis of the major lifestyle differences between these two groups of diatoms. The authors propose a tradeoff between retaining mixotrophy (flagellated cells of the Parmales) and building grazer defenses (silica cell walls of diatoms). This is an important conclusion as mixotrophy is a common feature of marine ecosystems and yet diatoms, one of the more successful phytoplankton, have lost mixotrophic capabilities. The manuscript is well-written with the key points clearly laid out. I greatly enjoyed reading it and learned a lot.

Thank you for your comments. We are glad to know that you agree with the main points of our work.

I have a few minor suggestions outlined below to clarify points.

1. Line 101. You should clarify that the 164 orthologous genes used for your analysis were identified in your study not based on previous work. I only realized this after reading

the methods section.

We appreciate your comment. We changed the text (L.103-105) to clarify this point as follows.

“We grouped the genes from the parmaleans (8 strains), diatoms (5 strains), and other stramenopiles (5 strains) and revealed 62,363 of orthologous groups (OGs).”

2. Figure 2. Initially, I was not sure what the numbers in Fig. 2a and 2d referred to. I assume they indicate the number of InterPro domains in category in each organism. If correct, this should be made explicit in the legend.

Thank you for your suggestion. The numbers in Fig 2a and 2d referred to the number of genes that have the InterPro domain in each organism. We added a caption (“#genes with IPR”) on the right of the table in Fig. 2a and Fig. 2d.

The following statement in the 2a legend should be clarified. “We manually clustered and selected the domains that appear to be involved in a specific process.” I think the authors could simply include (see methods).

We deleted this part from the Fig. 2a legend and added the following sentence in the Methods (L.514-515).

“Then, we manually selected and grouped the domains that are involved in specific biological processes.”

Legend for 2d includes the statement “InterPro domains that were not significant but were considered important are marked with an asterisk (*).” The domain that is not significantly enriched in the Parmales genomes should be removed from the figure.

Thank you for your suggestion. We deleted those domains from the Fig. 2d.

Fig. 2 has been changed as follows:

3. Figure 3. Fig 3a. An axis scale for this figure should be added. Fig. 3b legend. Need to indicate where full gene names can be found (as was done for 3c).

Thank you for your suggestion. We added an axis scale in Fig. 3a. As for Fig. 3b legend, we added the following text: “Gene names are abbreviated; full names and accessions can be found in Supplementary Data 6.”

In making this change, we also corrected an error in the figure. To be specific, the number

of Nitrate/Nitrite transporters has been corrected.

Fig. 3 has been changed as follows:

4. Figure 4 legend. The authors state “only important bootstrap values are noted.” Instead, the authors should state bootstrap values greater than X are shown, with X defined by the authors.

We appreciate your comment. We changed the figure to show bootstrap values. Bootstrap values greater than 50 are shown as circles now. We hope the change makes it easier to understand.

Fig. 4 has been changed as follows:

5. The authors should make available their “custom perl scripts”

Thank you for your suggestion. This script is a publicly available one and we cited the source in the revised version of our manuscript (L.431-433).

“The coverage was calculated from the resulting .sam file using

sam_len_cov_gc_insert.pl (<https://github.com/sujaikumar/assemblage>), which was also used to determine the GC content.”

6. Please add the parameters used for the HMM, MAFFT, and RAxML.

Thank you for your suggestion. We now clarified that we used the default parameters for these programs.

7. Please spell out the full name of the culture center.

Thank you for your suggestion. We spelled out the full name (L. 397-398).
“(NIES-2656; Microbial Culture Collection at the National Institute for Environmental Studies, Japan)”

8. Line 206-209 could benefit from a supplemental schematic.

We appreciate your comment. We added a new schematic figure (Supplemental Fig. 4).

Supplementary Fig. 4 is followed.

Reviewer #3 (Remarks to the Author):

Ban et al. generated seven parmalean genome assemblies and performed comparative genomic analysis between parmales and diatoms to dissect the genetic basis of adaptation in these two groups of organisms. The manuscript is straight forward and well-written. It has interesting and important results for the readers of Communications Biology.

However, I have some minor comments to the manuscript.

Thank you for your comments. We are happy to know that you found our work interesting and important.

Line 111. Not having InterPro domains does not indicate function of a protein is unknown. If the authors want to conclude that the lineage-specific genes of diatoms and parmaleans are functionally unknown, they should search the proteins in more datasets.

Thank you for your comment. We deleted this sentence.

Line 131. It would be better that the authors explain more why the enrichment of intracellular signaling pathways associate putative alternating life cycle stages in parmaleans.

Thank you for your comment. We emphasized this pathway may be used for the response to the external environment. We slightly modified the text (L.137-140).

From

“Enrichment of intracellular signalling pathways in parmaleans may be associated with their putative alternating life cycle stages (i.e., silicified/non-flagellated and naked/flagellated cell stages²).”

To

“Intercellular signalling in parmaleans may also be used to sense the external environment similarly to diatoms, and their enrichment may be related to performing the putative alternating life cycle stages (i.e., silicified/non-flagellated and naked/flagellated cell stages²) and/or to flagellar movement in response to the environment.”

Line 268. It is insufficient to conclude that silicanin genes in ciliate and dinoflagellate are derived from diatoms through HGT without phylogenetic analysis.

Thank you for your comment. We tried to generate phylogenetic trees of silicanin homologs, but could not obtain a robust result due to low bootstrap values at nodes throughout the whole tree. Therefore, we omitted the statement for the possibility of HGT and/or contamination. Nevertheless, we believe that our finding suggests that the common

ancestor of diatoms and pormaleans possessed silicanin genes. Accordingly, the text was modified as follow (L.277-292).

“Silicanin homologs have been reported in transcriptome data of other non-diatom eukaryotes such as the ciliate *Tiarina fusus* and the dictyochophyte *Rhizochromulina marina*⁵⁵. We also found 19 silicanin homologs from non-diatom eukaryote transcriptomes in the MMETSP database (15 sequences from *Tiarina fusus*, 2 from *Rhizochromulina marina*, 1 from the dinoflagellate *Durinskia baltica* and 1 from the dinoflagellate *Kryptoperidinium foliaceum*). Our finding of silicanin homologs in most of the analyzed pormaleans strongly suggests that the silicanin gene was already present in the diatom/Pormales common ancestor. Silicanins, like SITs, have undergone multiple gene duplications within the diatom lineage after the diatom/Pormales divergence. Interestingly, SIT and silicanin proteins were not found in any bolidomonad transcriptomes, which is consistent with their lack of silica plates.”

We also added a method related to this new analysis (L.540-544) as follows.

“Identification of silicanin homologs

To find silicanin homologs in MMETSP database⁸⁰, and our genomes, we used blastp search (Blast+ v2.10.1) with Sin1 gene of *Thalassiosira pseudonana* as query and default parameters. Among the hit sequences, we selected those with E-value < 1e-5 and > 300 aa and finally got 1991 silicanin homologs.”

REVIEWERS' COMMENTS:

Reviewer #1 (Remarks to the Author):

I am happy with the revisions made and responses provided by the authors in their rebuttal.

I have some minor suggestions concerning phrasing that I detail below:

Title: should it not be "the sister-group", unless there are other (e.g., uncultivated) algae that are more closely related to diatoms

Line 38: "undergone a loss of genes related to photosynthesis" or "have lost genes related to photosynthesis"

Line 39: "phago-mixotrophy" and "photoautotrophy"

Line 41: can you precise which nutrients? Iron and silica are also nutrients. An alternative phrasing would be "gene sets involved in nutrient uptake and metabolism, including iron and silica"

Line 63: "they are highly diverse, containing up to 10^5 species, and contribute extensively to marine primary production, performing up to 20% of total planetary photosynthesis"

Line 85: "their differential ecology" or "their different ecological features"

Line 122: "We noted that diatoms were enriched in cyclin domains and heat-shock transcription factor domains compared to Parmales, consistent with previous data that diatoms contain greater numbers of these proteins than other eukaryotes".

Line 124: precise "relative to Parmales"

Line 126: "sulfotransferases are enzymes" and "are implicated in programmed cell death"

Line 128: "relative to steps response processes"

Line 134: more formally, Phaeo has two planktonic and one benthic morphotype. We don't know enough about its environmental distribution (beyond it is an intertidal specialist) to know which morphotype predominates in nature

Line 137: Rephrase "Intercellular signalling pathways in parmaleans may also be used to sense the external environment similarly to as in diatoms. The enrichment of these pathways may relate to the putative alternating life cycle stages (i.e., silicified/non-flagellated and naked/flagellated cell stages) of Parmales, and/or to flagellar movement in response to the environment"

Line 175: Rephrase "To investigate the possibility that parmaleans can produce a flagellated cell, we searched for genes responsible for flagellar motility in the parmalean and diatom genomes and bolidomonad transcriptomes. The searched gene set included intraflagellar transport (IFT) subunit genes of the IFT-A complex (6 genes), IFT-B complex (15 genes), and BBSome (7 genes)." It would also be helpful to formally define the BBSome acronym here.

Line 185: "even in known sexually reproductive species"

Line 188: "IFT-A and BBSome genes", "IFT-B genes were partially detected"

Line 192: "the sperm of"

Line 193: Rephrase "Given the detection of the nearly-full set of flagellar genes in the parmaleans vs. the complete lack of IFT-A and BBSome and partial loss of IFT-B in the centric diatoms, it is possible that evolutionary pressure to maintain the flagellated stage is higher in parmaleans than in centric diatoms. This may be due to the presence of a frequent or prolonged flagellated stage in parmaleans, which is not expected for the sperm of centric diatoms."

Line 202: "Only one or no urea transporter gene was detected"

Line 206: Rephrase "Vacuolar nitrate transporters, which store nitrogen sources in the vacuole and are considered important for luxury nutrient uptake in diatoms, were absent from parmalean genomes."

Line 209: "although it remains to be determined if parmaleans utilise another vacuolar nitrate transport system that is not homologous to that of diatoms"

Line 215: "the production"

Line 217: Rephrase: "It should be noted that while parmalean genomes lacked NAD(P)H nitrate reductase, they retained a ferredoxin-nitrite reductase gene also found in diatoms that can perform the same activity. Likewise, parmaleans and diatoms share a carbamoyl phosphate synthetase enzyme that can function in lieu of carbamate kinase. The presence of multiple alternative pathways for these activities in diatoms, as opposed to only one in parmaleans, may enhance the efficiency of their nitrogen metabolism. Formamidase..."

Line 225: define what *Aureococcus* is, and its relevance to parmaleans

Line 231: "in photosynthesis and multiple other metabolic activities associated with phototrophs"

Line 253: you may note that most diatom plastocyanin genes (including that of *T. oceanica*) lack clear bipartite chloroplast targeting sequences, and their exact metabolic functions remain undefined

Line 257: are there multiple sequences from different *F. kerguelensis* strains? If not this is likely to be bacterial contamination.

Line 274: "The parmalean genomes contained low numbers (between 0 and 2 per species) of SIT homologues, compared to between 5 and 14 in diatoms"

Line 277: "the Sin1 and Sin2 genes"

Lines 285-287: *Kryptoperidinium* and *Durinskia* are dinoflagellates with diatom endosymbionts, so on the one hand the presence of SIT genes is unsurprising. On the other hand, it is surprising as dinoflagellate endosymbionts are not known to possess frustules. It would be worth elaborating on this in the text.

Line 300: Rephrase "Theoretically, phago-mixotrophs are less dependent on the uptake of inorganic nutrients than photoautotrophs. However, this advantage is traded off with an associated increase in metabolic costs for incorporating and maintaining the cellular components required for both autotrophy and phagotrophy."

Line 318: have you tried looking for homologues of the Parmalean mixotrophy genes in metaT data from Tara Oceans? This would be a great way to test if they are more expressed in oligotrophic/offshore stations.

Line 333: you should mention that osmotrophic mixotrophy is well known in diatoms, e.g. from glycerol feeding in *Phaeodactylum*; and also from secondarily non-photosynthetic and obligately osmotrophic species such as *Nitzschia putrida*

Line 371: "Diatoms also have important systemic impacts on marine iron usage"

Line 381: "parmalean groups"

Line 382: you could specify tropical open ocean regions, as coastal regions may be heavily fertilised by Aeolian and alluvial Fe deposition. Ustick 2021 Science is a good reference here.

Line 388: "the common ancestor"

Line 446: will the organelle genomes be released online? They will be a useful phylogenetic resource at least

Line 454: "contigs of the"

Line 458: "tBLASTn"- BLAST is an acronym and should be capitalised

Line 490: "the Silva database". In supporting Fig. 1 this is named "SILVA"- which is correct?

Line 494: "removed"

Line 502: "the 18 stramenopile genomes"?

Line 528: "from diatoms"

Line 544: "obtained 1,991 silicanin homologues"

Lines 891 and 896: "InterPro domains enriched in Diatom and Parmales genomes."

Line 899, also Supplementary Fig. 2: "predicted using a genome-scale tool developed by Burns et al."

Line 921: capitalise Parmales

Fig. 1. *T. columacea* is in a different font to the other species, correct.

Fig. 2. "Muliti-domains" is incorrectly spelt

Fig. 3. Should be "plastocyanin" (lowercase) and "independently lost"

Fig. 4. The Parmales clade has been expanded relative to the other branches. This has a phylogenetic meaning, tied to both the speciation rate and subselection of branches shown on the tree. Just say in

the legend that this clade has been manually expanded to permit legibility.

Fig. 5. "Specialization as photoautotrophs"; "Silicate transporters".

Supplementary Line 29: "bolidomonad transcriptomes"

Supplementary Line 32: "obtained from the genome sequences of cultured members"

Supplementary Line 35: "diatom MAGs"

Supplementary Line 38: "supports the idea that diatoms are universally not phagotrophic"

Supplementary Line 39: "Some diatoms have N₂-fixing cyanobacterial endosymbionts, or non-photosynthetic symbionts termed 'spheroid bodies'"

Supplementary Line 42: "To check whether these diatoms possess the capacity for phagotrophy signals, we examined a diatom with endosymbionts"

Supplementary Line 48: "land plants"

Supplementary Line 81: replace CCA with "biophysical carbon concentration mechanisms". What are the distributions of PEP carboxylase, PEP carboxykinase, and malic enzymes (i.e., biochemical CCM enzymes) in Parmales?

Supplementary Line 90: you should really cite Rio Bartulos 2018, where this phenomenon was first described

Supplementary Line 106: was EDA detectable in any non-diatom stramenopile MMETSP transcriptomes beyond Parmales? What about in pelagophytes and dictyochophytes?

Fig. S1: just "Genomic strains used in this study"

Supplementary Line 181: "Stars indicate the sequenced genomic strains, and red circles isolated bolidomonad strains."

Supplementary Figs. 3, 7: explain how localisations were predicted (briefly) for readers who have not consulted the Methods

Supplementary Fig. 4: "pathways in N metabolism", "arrows show"

Supplementary Fig. 5: taxonomic labels are inconsistent. Could be "Other Stramenopiles"; "Rhizaria"; "Haptophytes"; "Glaucophytes"; "Viruses". T columacea font is again a bit strange. The star really shows the monophyly of diatoms, parmales and dictyochophytes- the Minchinia sequence is likely a diatom contaminant. Seriola lalandi lacks a taxonomic assignation

Supplementary Fig. 6: should be "taxon". Single quotation marks needed around 'Scaly parma'

Supplementary Line 215: "the total"

Supplementary Line 265: "targeting predictions"

Supplementary Data Legends text: please place a copy of this in the supporting data workbook (a "contents" tab for user navigation

Supplementary Data 1: "stress response"

Supplementary Data 2: "receptor" not "receptor", "Ca signaling" is more formally "Ca²⁺ signalling"

Supplementary Data 4: capitalise "Bolidomonas"

Supplementary Data 11: add species/ strain names

Reviewer #3 (Remarks to the Author):

My comments have been addressed satisfactorily. I suggest to accept this manuscript.

REVIEWERS' COMMENTS:

Reviewer #1 (Remarks to the Author):

I am happy with the revisions made and responses provided by the authors in their rebuttal.

I have some minor suggestions concerning phrasing that I detail below:

Thank you again for your valuable comments. We have carefully considered your suggestions and made the necessary changes. We answer your specific comments below.

Title: should it not be “the sister-group”, unless there are other (e.g., uncultivated) algae that are more closely related to diatoms

Thank you for your comment. As you pointed out, we have made the necessary corrections.

Line 38: “undergone a loss of genes related to photosynthesis” or “have lost genes related to photosynthesis”

Thank you for suggesting the revision. We took the latter one and we have made the correction (L. 38).

Line 39: “phago-mixotrophy” and “photoautotrophy”

Thank you for proposing the revision. We appreciate your help (L. 39).

Line 41: can you precise which nutrients? Iron and silica are also nutrients. An alternative phrasing would be “gene sets involved in nutrient uptake and metabolism, including iron and silica”

Thank you for your comment. We modified the text according to your suggestion (L. 40-41).

Line 63: “they are highly diverse, containing up to 10^5 species, and contribute extensively to marine primary production, performing up to 20% of total planetary photosynthesis”

Thank you for proposing the revision. We followed your suggestion (L. 64-66).

Line 85: “their differential ecology” or “their different ecological features”

Thank you for proposing the revision. We rephrased “their ecological features” (L. 87).

Line 122: “We noted that diatoms were enriched in cyclin domains and heat-shock transcription factor domains compared to Parmales, consistent with previous data that diatoms contain greater numbers of these proteins than other eukaryotes”.

Thank you for proposing the revision. We followed your suggestion (L. 121-124).

Line 124: precise “relative to Parmales”

Thank you for proposing the revision. We followed your suggestion (L. 127).

Line 126: “sulfotransferases are enzymes” and “are implicated in programmed cell death”

Thank you for proposing the revision. We followed your suggestion (L. 128-129).

Line 128: “relative to steps response processes”

Thank you for proposing the revision. We rephrased “involved in the stress response process in diatoms” (L. 131).

Line 134: more formally, Phaeo has two planktonic and one benthic morphotype. We don't know enough about its environmental distribution (beyond it is an intertidal specialist) to know which morphotype predominates in nature

Thank you for your comment. Regarding your feedback, we have clarified that we are discussing the benthic morphotype (L. 136-137).

Line 137: Rephrase “Intercellular signalling pathways in parmaleans may also be used to sense the external environment similarly to as in diatoms. The enrichment of these pathways may relate to the putative alternating life cycle stages (i.e., silicified/non-flagellated and

naked/flagellated cell stages) of Parmales, and/or to flagellar movement in response to the environment”

Thank you for proposing the revision. We followed your suggestion (L. 140-144).

Line 175: Rephrase “To investigate the possibility that parmaleans can produce a flagellated cell, we searched for genes responsible for flagellar motility in the parmalean and diatom genomes and bolidomonad transcriptomes. The searched gene set included intraflagellar transport (IFT) subunit genes of the IFT-A complex (6 genes), IFT-B complex (15 genes), and BBSome (7 genes).” It would also be helpful to formally define the BBSome acronym here.

Thank you for proposing the revision. We followed your suggestion (L. 178-181). We also define the BBSome (Bardet–Biedl Syndrome proteins; L. 180).

Line 185: “even in known sexually reproductive species”

Thank you for proposing the revision. We followed your suggestion (L. 188-189).

Line 188: “IFT-A and BBSome genes”, “IFT-B genes were partially detected”

Thank you for proposing the revision. We followed your suggestion (L. 192).

Line 192: “the sperm of”

Thank you for your correction (L. 196, 201)

Line 193: Rephrase “Given the detection of the nearly-full set of flagellar genes in the parmaleans vs. the complete lack of IFT-A and BBSome and partial loss of IFT-B in the centric diatoms, it is possible that evolutionary pressure to maintain the flagellated stage is higher in parmaleans than in centric diatoms. This may be due to the presence of a frequent or prolonged flagellated stage in parmaleans, which is not expected for the sperm of centric diatoms.”

Thank you for proposing the revision. We followed your suggestion (L. 197-201).

Line 202: “Only one or no urea transporter gene was detected”

Thank you for your correction (L. 206-207).

Line 206: Rephrase “Vacuolar nitrate transporters, which store nitrogen sources in the vacuole and are considered important for luxury nutrient uptake in diatoms, were absent from parmalean genomes.”

Thank you for proposing the revision. We followed your suggestion (L. 209-212).

Line 209: “although it remains to be determined if parmaleans utilise another vacuolar nitrate transport system that is not homologous to that of diatoms”

Thank you for proposing the revision. We followed your suggestion (L. 212-215).

Line 215: “the production”

Thank you for your correction (L. 220).

Line 217: Rephrase: “It should be noted that while parmalean genomes lacked NAD(P)H

nitrate reductase, they retained a ferredoxin-nitrite reductase gene also found in diatoms that can perform the same activity. Likewise, parmaleans and diatoms share a carbamoyl phosphate synthetase enzyme that can function in lieu of carbamate kinase. The presence of multiple alternative pathways for these activities in diatoms, as opposed to only one in parmaleans, may enhance the efficiency of their nitrogen metabolism. Formamidase...”
Thank you for proposing the revision. We followed your suggestion (L. 222-225).

Line 225: define what *Aureococcus* is, and its relevance to parmaleans
Thank you for your comment. We have revised the text based on the comment (L. 234-235).

Line 231: “in photosynthesis and multiple other metabolic activities associated with phototrophs”
Thank you for proposing the revision. We followed your suggestion (L. 241-242).

Line 253: you may note that most diatom plastocyanin genes (including that of *T. oceanica*) lack clear bipartite chloroplast targeting sequences, and their exact metabolic functions remain undefined.
Thank you for your comment. We were unable to find the specific paper describing your point (i.e., lack of chloroplast targeting signals). We agree with you that there are still unresolved questions regarding the precise function of plastocyanin genes of diatoms. However, some indirect evidence suggests their involvement in the photosynthetic process, as demonstrated by Hippmann et al. (2017) in PLOS ONE. We decided not to discuss this lack of evidence for detailed functionality of plastocyanin, although we believe further research is needed to elucidate the actual mechanisms through which these genes operate to keep the conciseness.

Line 257: are there multiple sequences from different *F. kerguelensis* strains? If not this is likely to be bacterial contamination.
Thank you for your comments. There are two strains of *F. kerguelensis* sequenced in the MMETSP project, and the plastocyanin gene grouped with the bacteria is derived from only one of them, so contamination is possible. But assessing this is beyond the scope of our study, so we will not discuss it.
Related to this, we changed the text to emphasize that there are multiple plastocyanin genes derived from *F. kerguelensis*, and one of them is grouped with the bacteria (L. 266).

Line 274: “The parmalean genomes contained low numbers (between 0 and 2 per species) of SIT homologues, compared to between 5 and 14 in diatoms”

Thank you for your correction (L. 283-285).

Line 277: “the Sin1 and Sin2 genes”

Thank you for your correction (L. 287).

Lines 285-287: Kryptoperidinium and Durinskia are dinoflagellates with diatom endosymbionts, so on the one hand the presence of SIT genes is unsurprising. On the other hand, it is surprising as dinotom endosymbionts are not known to possess frustules. It would be worth elaborating on this in the text.

Thank you for your comments. We add the text to the manuscript to address these facts (L.292-294).

Line 300: Rephrase “Theoretically, phago-mixotrophs are less dependent on the uptake of inorganic nutrients than photoautotrophs. However, this advantage is traded off with an associated increase in metabolic costs for incorporating and maintaining the cellular components required for both autotrophy and phagotrophy.”

Thank you for proposing the revision. We rephrase the text as followed (L. 305-307).

“”

Therefore, phago-mixotrophs is considered less dependent on the uptake of inorganic nutrients than photoautotrophs., However but this advantage is traded off with an associated increase in metabolic costs for incorporating and maintaining the cellular components required for both autotrophy and phagotrophy.

“”

Line 318: have you tried looking for homologues of the Parmalean mixotrophy genes in metaT data from Tara Oceans? This would be a great way to test if they are more expressed in oligotrophic/ offshore stations.

While the answer is no for now, it is indeed a valuable topic, and we would like to explore it in future research. Thank you for your nice suggestion.

Line 333: you should mention that osmotrophic mixotrophy is well known in diatoms, e.g. from glycerol feeding in Phaeodactylum; and also from secondarily non-photosynthetic and obligately osmotrophic species such as Nitzschia putrida

Thank you for your comment. We agree with your point and have accordingly made

changes to the content (L. 338-341).

Line 371: “Diatoms also have important systemic impacts on marine iron usage”

Thank you for proposing the revision. We followed your suggestion (L. 376).

Line 381: “parmalean groups”

Thank you for your correction (L. 385).

Line 382: you could specify tropical open ocean regions, as coastal regions may be heavily fertilised by Aeolian and alluvial Fe deposition. Ustick 2021 Science is a good reference here.

Thank you for your thoughtful comment. We modified “tropical” to “tropical open ocean”.

Line 388: “ the common ancestor”

Thank you for your correction (L. 394).

Line 446: will the organelle genomes be released online? They will be a useful phylogenetic resource at least

Thank you for your suggestions. We are currently advancing our research about it, so we will not make it public at this time. We hope you understand our standpoint.

Line 454: “contigs of the”

Thank you for your correction (L. 459).

Line 458: “tBLASTn”- BLAST is an acronym and should be capitalised

Thank you for your correction (L. 463).

Line 490: “the Silva database”. In supporting Fig. 1 this is named “SILVA”- which is correct?

Thank you for your comment. “SILVA” is correct (L. 495).

Line 494: “removed”

Thank you for your correction (L. 498).

Line 502: ”the 18 stramenopile genomes”?

Thank you for your correction (L. 509).

Line 528: “from diatoms”

Thank you for your correction (L. 536).

Line 544: “obtained 1,991 silicanin homologues”

Thank you for your correction (L. 552).

Lines 891 and 896: “InterPro domains enriched in Diatom and Parmales genomes.”

Thank you for your suggestion (L. 911, 918).

Line 899, also Supplementary Fig. 2: “predicted using a genome-scale tool developed by Burns et al.”

Thank you for proposing the revision. We followed your suggestion (L. 921).

Line 921: capitalise Parmales

Thank you for your correction (L. 946).

Fig. 1. *T. columacea* is in a different font to the other species, correct.

Thank you for your correction.

Fig. 2. “Muliti-domains” is incorrectly spelt

Thank you for your correction.

Fig. 3. Should be “plastocyanin” (lowercase) and “independently lost”

Thank you for your correction.

Fig. 4. The Parmales clade has been expanded relative to the other branches. This has a phylogenetic meaning, tied to both the speciation rate and subselection of branches shown on the tree. Just say in the legend that this clade has been manually expanded to permit legibility.

Thank you for your comment. We added the explanation to the legend.

Fig. 5. “Specialization as photoautotrophs”; “Silicate transporters”.

Thank you for your correction.

Supplementary Line 29: “bolidomonad transcriptomes”

Thank you for your correction.

Supplementary Line 32: “obtained from the genome sequences of cultured members”

Thank you for proposing the revision. We followed your suggestion (L. 32).

Supplementary Line 35: “diatom MAGs”

Thank you for your correction (L. 36).

Supplementary Line 38: “supports the idea that diatoms are universally not phagotrophic”

Thank you for proposing the revision. We followed your suggestion (L. 38-39).

Supplementary Line 39: “Some diatoms have N₂-fixing cyanobacterial endosymbionts, or non-photosynthetic symbionts termed ‘spheroid bodies’”

Thank you for proposing the revision. We followed your suggestion (L. 40-42).

Supplementary Line 42: “To check whether these diatoms possess the capacity for phagotrophy signals, we examined a diatom with endosymbionts”

Thank you for proposing the revision. We followed your suggestion (L. 43-44).

Supplementary Line 48: “land plants”

Thank you for your correction (L. 51).

Supplementary Line 81: replace CCA with “biophysical carbon concentration mechanisms”. What are the distributions of PEP carboxylase, PEP carboxykinase, and malic enzymes (i.e., biochemical CCM enzymes) in Parmales?

Thank you for your comment. We revised the manuscript (L. 84).

Regarding biochemical CCM enzymes (C₄ pathway enzymes), the reasons for not discussing their distribution in this manuscript have already been described in our previous reply to your previous comments. For your information, our reply was:

“However, it is difficult to prove the existence of the C₄ pathway in parmaleans based on protein localization predictions alone (like in other previous studies, c.f., Kroth et al., PLoS One, 2008), and experimental evidence is essential (e.g., Yu et al. 2022). Furthermore, the localizations of genes involved in the C₄ shunt (PEPC/MDH/PPDK) were not predicted to be the same within diatoms (Supplemental Fig. 6), indicating that results on P. tricornutum from Yu et al. may not be generalized across diverse diatoms

and could not be attributed to their ecological success. Since we focused on delineating differences between diatoms and parmaleans and discussing this topic would be still highly speculative, we decide not to discuss this topic.”

Supplementary Line 90: you should really cite Rio Bartulos 2018, where this phenomenon was first described

Thank you for your comment. We newly cited this paper (L. 94).

Supplementary Line 106: was EDA detectable in any non-diatom stramenopile MMETSP transcriptomes beyond Parmales? What about in pelgophytes and dictyochophytes?

Thank you for your comment. We haven't investigated it, but the topic is interesting, and we would like to explore it in future research.

Fig. S1: just “Genomic strains used in this study”

Thank you for your correction.

Supplementary Line 181: “Stars indicate the sequenced genomic strains, and red circles isolated bolidomonad strains.”

Thank you for proposing the revision. We followed your suggestion (L. 195-196).

Supplementary Figs. 3, 7: explain how localisations were predicted (briefly) for readers who have not consulted the Methods

Thank you for your suggestion. We added the text to the legends (L. 202-203 and 226-227).

Supplementary Fig. 4: “pathways in N metabolism”, “arrows show”

Thank you for proposing the revision. We followed your suggestion (L. 207-208).

Supplementary Fig. 5: taxonomic labels are inconsistent. Could be “Other Stramenopiles”; “Rhizaria”; “Haptophytes”; “Glaucophytes”; “Viruses”. T columacea font is again a bit strange. The star really shows the monophyly of diatoms, parmales and dictyochophytes- the Minchinia sequence is likely a diatom contaminant. Seriola lalandi lacks a taxonomic assignation

Thank you for your correction.

Supplementary Fig. 6: should be “taxon”. Single quotation marks needed around ‘Scaly

parma'

Thank you for your correction.

Supplementary Line 215: "the total"

Thank you for your correction.

Supplementary Line 265: "targeting predictions"

Thank you for your correction.

Supplementary Data Legends text: please place a copy of this in the supporting data workbook (a "contents" tab for user navigation

Thank you for your suggestion. We newly added a new tab for navigation.

Supplementary Data 1: "stress response"

Thank you for your correction.

Supplementary Data 2: "receptor" not "receptor", "Ca singling" is more formally "Ca²⁺ signalling"

Thank you for your correction.

Supplementary Data 4: capitalise "Bolidomonas"

Thank you for your correction.

Supplementary Data 11: add species/ strain names

Thank you for your suggestion and we followed it.

Reviewer #3 (Remarks to the Author):

My comments have been addressed satisfactorily. I suggest to accept this manuscript.

Thank you for your positive feedback and recommendation for acceptance. We are grateful for your support and are delighted to know that our work has been well received.